Microbiology **Spectrum**

# Caffeic Acid Phenethyl Ester (CAPE) Inhibits Cross-Kingdom Biofilm Formation of *Streptococcus mutans* and *Candida albicans*

Wumeng Yin,[a] Zhong Zhang,[a] Xinxing Shuai,[a] Xuedong Zhou,[a] Derong Yin[a]

[a]State Key Laboratory of Oral Diseases, National Clinical Research Center for Oral Diseases, West China Hospital of Stomatology, Sichuan University, Chengdu, PR China

**ABSTRACT** *Streptococcus mutans* and *Candida albicans* exhibit strong cariogenicity through cross-kingdom biofilm formation during the pathogenesis of dental caries. Caffeic acid phenethyl ester (CAPE), a natural compound, has potential antimicrobial effects on each species individually, but there are no reports of its effect on this dual-species biofilm. This study aimed to explore the effects of CAPE on cariogenic biofilm formation by *S. mutans* and *C. albicans* and the related mechanisms. The effect of CAPE on planktonic cell growth was investigated, and crystal violet staining, the anthrone-sulfuric acid assay and confocal laser scanning microscopy were used to evaluate biofilm formation. The structures of the formed biofilms were observed using scanning electron microscopy. To explain the antimicrobial effect of CAPE, quantitative real-time PCR (qRT–PCR) was applied to monitor the relative expression levels of cariogenic genes. Finally, the biocompatibility of CAPE in human oral keratinocytes (HOKs) was evaluated using the CCK-8 assay. The results showed that CAPE suppressed the growth, biofilm formation and extracellular polysaccharides (EPS) synthesis of *C. albicans* and *S. mutans* in the coculture system of the two species. The expression of the *gtf* gene was also suppressed by CAPE. The efficacy of CAPE was concentration dependent, and the compound exhibited acceptable biocompatibility. Our research lays the foundation for further study of the application of the natural compound CAPE as a potential antimicrobial agent to control dental caries-associated cross-kingdom biofilms.

**IMPORTANCE** Severe dental caries is a multimicrobial infectious disease that is strongly induced by the cross-kingdom biofilm formed by *S. mutans* and *C. albicans*. This study aimed to investigate the potential of caffeic acid phenethyl ester (CAPE) as a natural product in the prevention of severe caries. This study clarified the inhibitory effect of CAPE on cariogenic biofilm formation and the control of cariogenic genes. It deepens our understanding of the synergistic cariogenic effect of *S. mutans* and *C. albicans* and provides a new perspective for the prevention and control of dental caries with CAPE. These findings may contribute to the development of CAPE as a promising antimicrobial agent targeting this caries-related cross-kingdom biofilm.

**KEYWORDS** caffeic acid phenethyl ester, *Candida albicans*, Streptococcus mutans, cross-kingdom biofilm, dental caries

Address correspondence to Xuedong Zhou, zhouxd@scu.edu.cn, or Derong Yin, derongyin@scu.edu.cn.

The authors declare no conflict of interest.

Dental caries is one of the most common tooth diseases in the worldwide; the majority of adults and 60–90% of children suffer from dental caries, according to the WHO (1). Caries is a kind of chronic processive devastation of dental hard tissue caused by microbe-associated factors. The typical characteristic of caries is the cariogenic biofilm (dental plaque) formed on tooth surfaces via the aggregation of cariogenic pathogens, the bacteria in which metabolize carbohydrates to produce acids, leading to demineralization of dental hard tissue (2, 3). Considered a key pathogen, *Streptococcus mutans* is involved in the occurrence and advancement of caries. Via the action of a secreted glycosyltransferase (Gtf), *S. mutans* ferments sugar to produce water-insoluble extracellular polysaccharides (EPS), mediating the adhesion of planktonic bacteria and promoting biofilm formation, which plays an essential role in caries progression (4, 5).

Studies at the transcriptional level have indicated that the assembly and maturation of microbial communities in biofilms are complex (6). Interestingly, coinfection characterized by bacterial-fungal symbiosis in biofilms has offered a deeper understanding of the progression of dental caries (7, 8). According to some reports, *Candida albicans* can synergistically enhance the infection caused by *S. mutans* (9–11). Specifically, *C. albicans* promotes the synthesis of EPS, contributing to increasing the biomass of *S. mutans* biofilms and the formation of cross-kingdom biofilms (8), which aggravates oral infections, especially in early childhood caries (ECC) (7, 12). Therefore, inhibition of cross-kingdom biofilm formation is likely an effective method for the intervention of dental caries.

However, most traditional clinical drugs are based on individual antibacterial or antifungal agents (13). Studies have shown that the intervention of cross-kingdom biofilms produced by *C. albicans* and *S. mutans* is challenging (7). The application of various antimicrobial agents in currently used clinical drugs have led to the emergence of bacterial resistance (14, 15). Some of these compounds, including chlorhexidine (CHX), even cause homeostatic imbalance in the oral microenvironment, resulting in the recurrence of dental caries (16). Cytotoxicity and dental staining also cause issues in the use of some antimicrobial agents for routine oral hygiene (17). To overcome these limitations of traditional antimicrobials, it is of great significance to develop an alternative product that enables efficient inhibition of cariogenicity as well as enhances the oral microecological balance.

Natural compounds, as a novel and more attractive option with good biocompatibility, are considered to have preventive potential for dental caries (18–20). Caffeic acid phenethyl ester (CAPE), a natural compound extracted from propolis, has been recognized for its biological and pharmacological properties, including its anti-inflammatory, antioxidant, antibacterial, immunomodulatory, anticancer, and wound healing and repair properties (21, 22).

Based on the pleiotropic effects of CAPE, its potential utility in the prevention of oral diseases has been described in some reports, in which it displayed antibacterial activity against *S. mutans* (5) and antifungal activity against *C. albicans* (23, 24). Although related studies have independently expressed the antimicrobial effect and associated mechanism of action of CAPE against *S. mutans* and *C. albicans*, there has been no exploration of the inhibitory effect on the state of both species. In general, the effects of CAPE on the cariogenic virulence of cross-kingdom biofilms formed by *C. albicans* and *S. mutans* are still unclear, which prompted us to study this topic further.

In our research, *S. mutans* and *C. albicans* were applied to construct single- or multispecies biofilm models, and the influence of CAPE on pathogenic biofilm formation was measured. The results of our study provide evidence for the promising application of CAPE as an innovative antibiofilm agent in the prevention and control of dental caries.

## RESULTS

**CAPE inhibits planktonic growth of *S. mutans* and *C. albicans*.** Based on the susceptibility assay, the MIC of CAPE for the *S. mutans* UA159 strain was 160 $\mu$g/mL. Simultaneously, CAPE suppressed the growth of the *C. albicans* SC5314 strain at concentrations ranging from 80 to 160 $\mu$g/mL. The MIC of CAPE for *C. albicans* was determined to be 80 $\mu$g/mL in accordance with the definition. DMSO at 5% (vol/vol) as the solvent control had no effect on bacterial and fungal growth. The MIC of CHX ranged from $4 \times 10^3$ to $8 \times 10^3$ mg/mL for both strains as the positive control, which was significantly higher than the MIC of CAPE. Thus, 20, 40, 80, and 160 $\mu$g/mL CAPE were selected as the experimental groups, and the solvent group was selected as the control group in the present study.

In further experimental confirmation, CAPE at increasing concentrations displayed a significant bacteriostatic influence on *S. mutans*, *C. albicans* and coculture strains (Fig. 1A). To examine the corresponding variations of colony numbers in *S. mutans* and *C. albicans* under inhibition by CAPE, the CFU of the planktonic cultures were determined (Fig. 1B and C). Similarly, the numbers of planktonic cells of both species declined with increasing CAPE

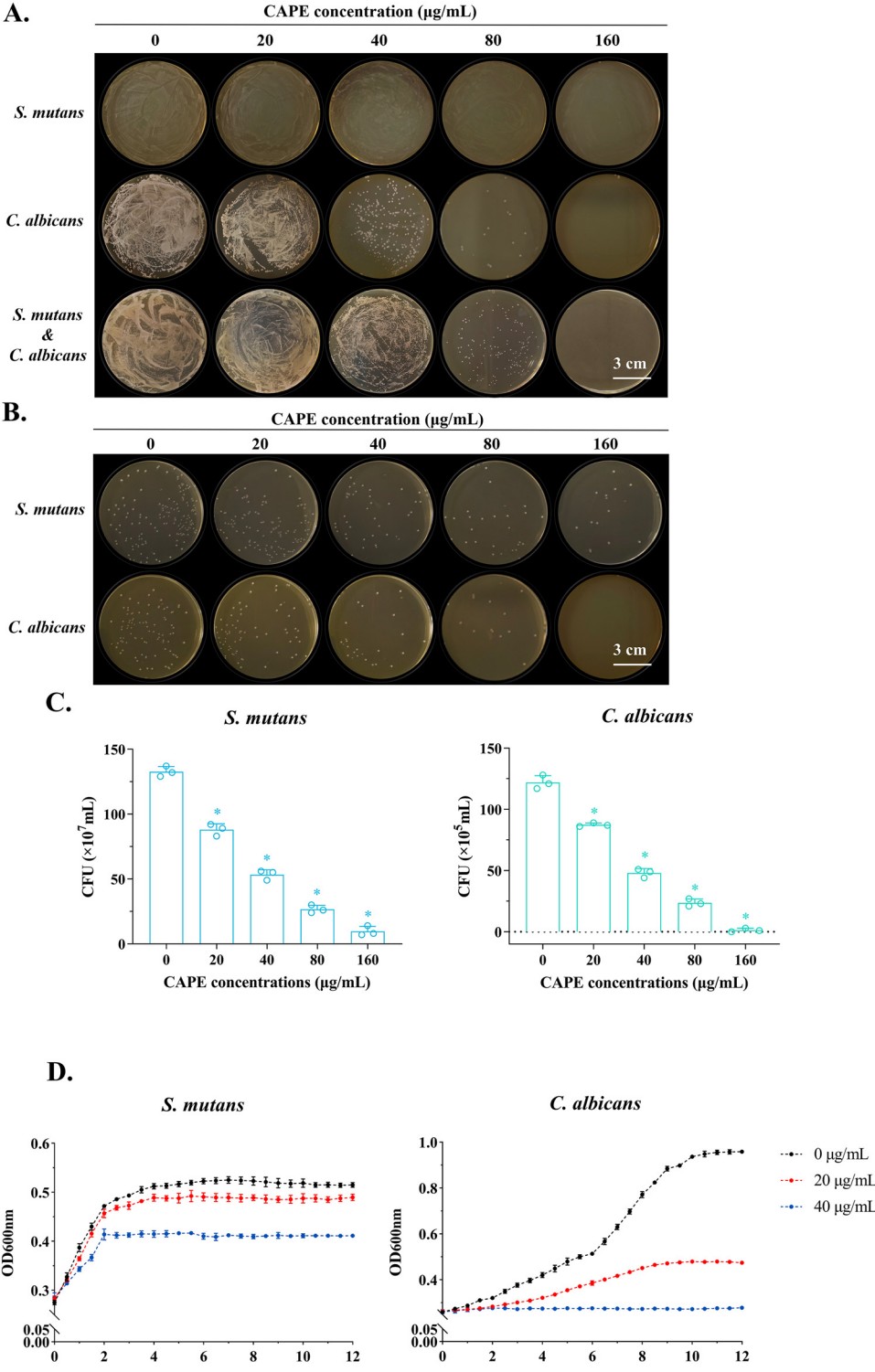

**FIG 1** Inhibitory effects of CAPE on the planktonic growth of *S. mutans* and *C. albicans*. (A) Images of the plate coating assay applied to supplement the MIC measurement. (B) Images of CFU counts for *S. mutans* and *C. albicans* colonies. (C) Quantitative measurement of colony numbers in CFU counting images. (D) Growth curves based on absorbance measurement for CAPE in *S. mutans* and *C. albicans*. The values are expressed as the mean ± standard deviation (SD) for at least three independent trials (*, $P < 0.05$).

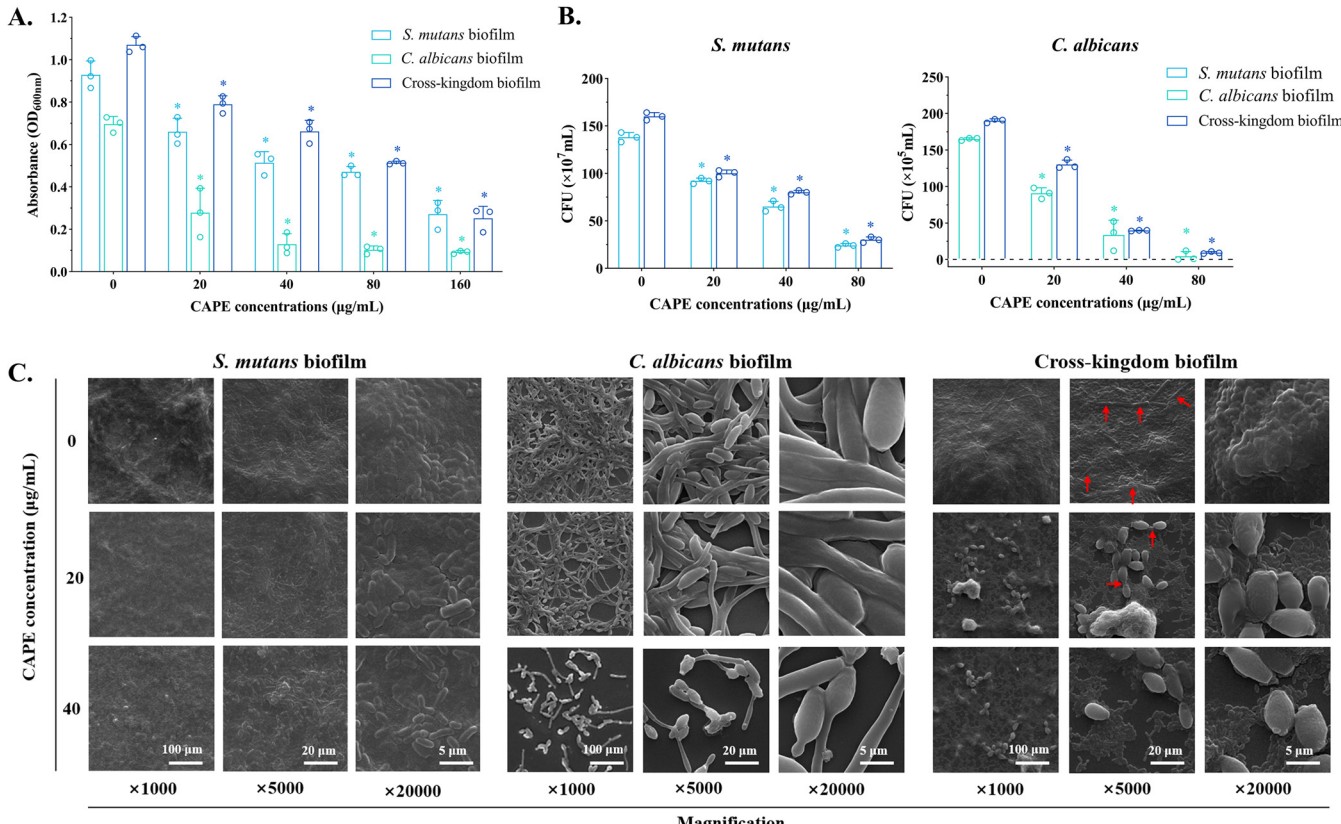

**FIG 2** Inhibitory effects of CAPE against biofilm formation. (A) Effect of CAPE on biofilm formation, as determined by the crystal violet staining assay. (B) Results of CFU counting to measure biofilm formation. (C) Surface morphology of biofilms produced on glass coverslips affected by CAPE. The microimages were taken at ×1,000, ×5,000 and ×20,000 magnifications by scanning electron microscopy. The red arrows indicate the formation of *C. albicans* as yeast or hyphae in the cross-kingdom biofilms. The values represent the mean ± standard deviation (SD) of at least three independent trials (*, $P < 0.05$).

concentration. Furthermore, CAPE showed observable inhibitory activity on the microbial growth patterns of *S. mutans* and *C. albicans* (Fig. 1D). At 40 $\mu$g/mL (1/4 MIC), CAPE caused an approximately 2-fold reduction in the OD600 of *S. mutans*. With regard to the growth of *C. albicans*, the optical density in the system indicated that *C. albicans* hardly proliferated and grew sluggishly with 1/2 MIC CAPE. The growth and proliferation of both species were suppressed with CAPE treatment.

**CAPE inhibits formation of *S. mutans*, *C. albicans* and dual-species biofilms.** The results of the crystal violet staining assay exhibited an effective reduction in the biomass of biofilms in all CAPE-treated groups (Fig. 2A). The higher the concentration of CAPE was, the stronger the inhibitory effect on single-species and cross-kingdom biofilm formation. As observed in all groups, *S. mutans* promoted *C. albicans* biofilm formation, and vice versa. Furthermore, the colony numbers of the biofilms are displayed in Fig. 2B. The bacterial and fungal cell counts decreased in the presence of CAPE with the same trend both in single-species and cross-kingdom biofilms. Compared with single-species biofilms, the colonies formed in cross-kingdom biofilms showed a slight increase in all groups, which was consistent with the outcomes of CFU counting. This result suggested a synergistic effect between *C. albicans* and *S. mutans* in cross-kingdom biofilm formation that was affected by CAPE.

The SEM photographs of biofilms are presented in Fig. 2C. The distribution of *C. albicans* and *S. mutans* cells inside single-species and cross-kingdom biofilms is shown. For *S. mutans* biofilms, clusters of bacterial cells were covered with a high density of the water-insoluble EPS in the control group, while the CAPE-treated groups showed some collapse of the biofilm, with fewer cells and fewer structured stacks. For *C. albicans*, thinner structured biofilms were formed with CAPE compared with the control group. Moreover, 40 $\mu$g/mL CAPE led to morphological deformation in the *C. albicans*

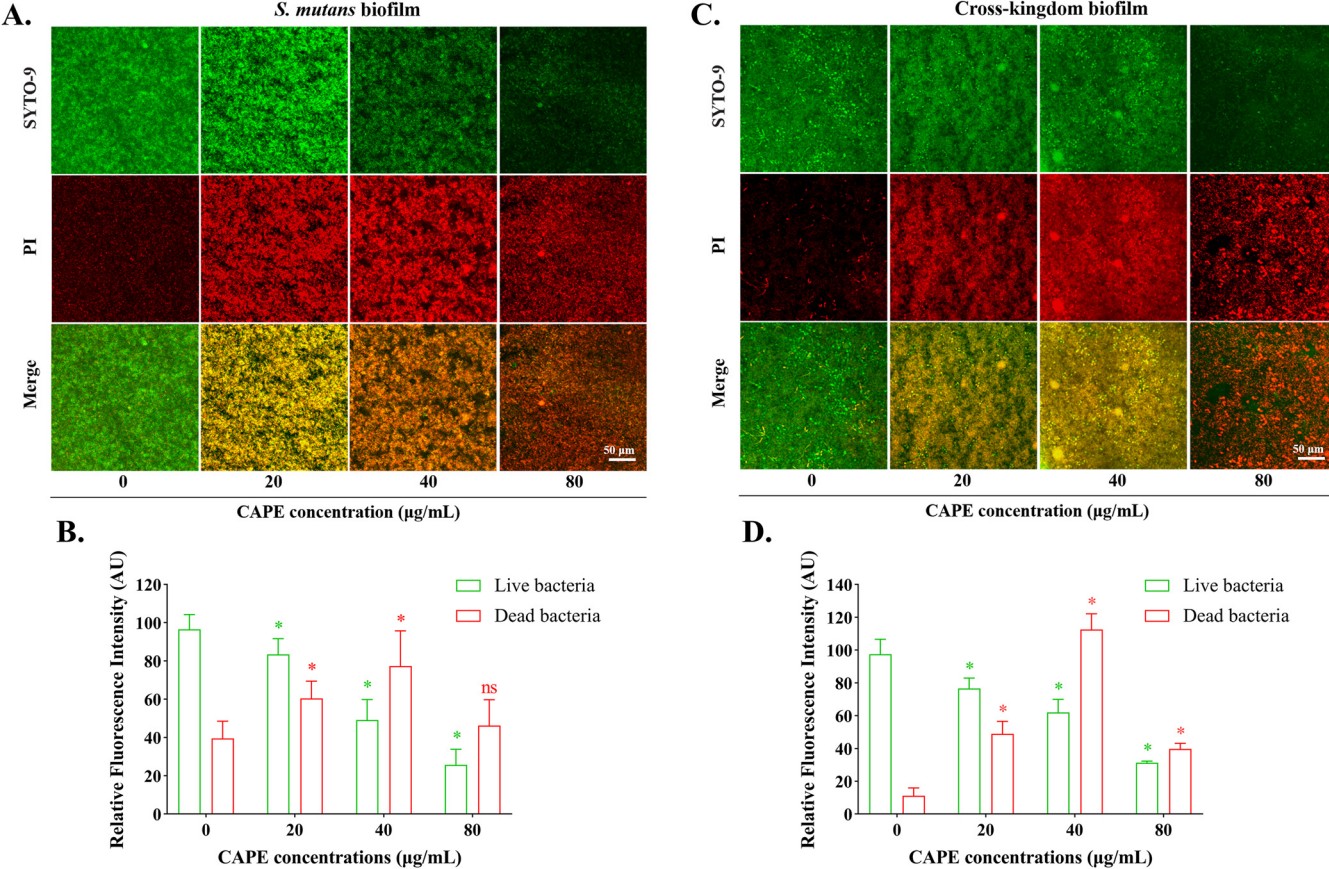

**FIG 3** Bacterial viability in *S. mutans* and cross-kingdom biofilms with CAPE treatment. (A, C) Images of the live/dead staining of *S. mutans* and cross-kingdom biofilms. The green color (SYTO-9) and the red color (PI) represent live and dead bacteria, respectively. (B, D) Biomass of live and dead bacteria in *S. mutans* and the cross-kingdom biofilms presented as the relative fluorescence intensity. Statistical data are expressed as the mean ± standard deviation (SD) from at least three independent experiments (*, $P < 0.05$).

biofilm. For the cross-kingdom biofilms, it was obvious that *C. albicans* colonies were closely surrounded by *S. mutans* biofilms that gathered into stacks in the absence of CAPE. With CAPE treatment, only a few *C. albicans* cells were attached to the cross-kingdom biofilms, which had a lower density of *S. mutans* cells and lower EPS levels.

**CLSM shows inhibitor effect of CAPE on growth and biofilm formation.** The live/dead bacterial viability assays performed using CLSM showed a marked decrease in *S. mutans* and cross-kingdom biofilms, as shown in Fig. 3. The SYTO-9 dye stained the live bacteria green, and PI stained the dead bacteria red.

Among the *S. mutans* biofilms exhibited in Fig. 3A, the biofilms in the 0 μg/mL CAPE group (the control group) were mostly green and thick. However, in the other groups, the thickness of biofilms and the green area gradually declined with increasing concentrations of CAPE, while the red area constantly increased. As shown in the graph of the fluorescence intensity (Fig. 3B), the biomass of *S. mutans* biofilms, indicated by the green fluorescence intensity, gradually decreased from 0 μg/mL to 80 μg/mL CAPE, whereas the red fluorescence intensity increased, except for 80 μg/mL CAPE. Since there was less biofilm formation at a CAPE concentration of 80 μg/mL, the proportion of dead bacteria from biofilms in the 40 μg/mL group was significantly higher than that at 80 μg/mL, which is consistent with the result of the biofilm formation assay.

Similarly, for the cross-kingdom biofilms shown in Fig. 3C, it is clear that the observable reduction in biofilm accumulation correlated with increasing CAPE concentrations. The increasing concentration of CAPE also resulted in a reduced proportion of living bacteria and an increased proportion of dead bacteria, as shown in Fig. 3D, and the red fluorescence intensity at a CAPE concentration of 80 μg/mL was lower than that at

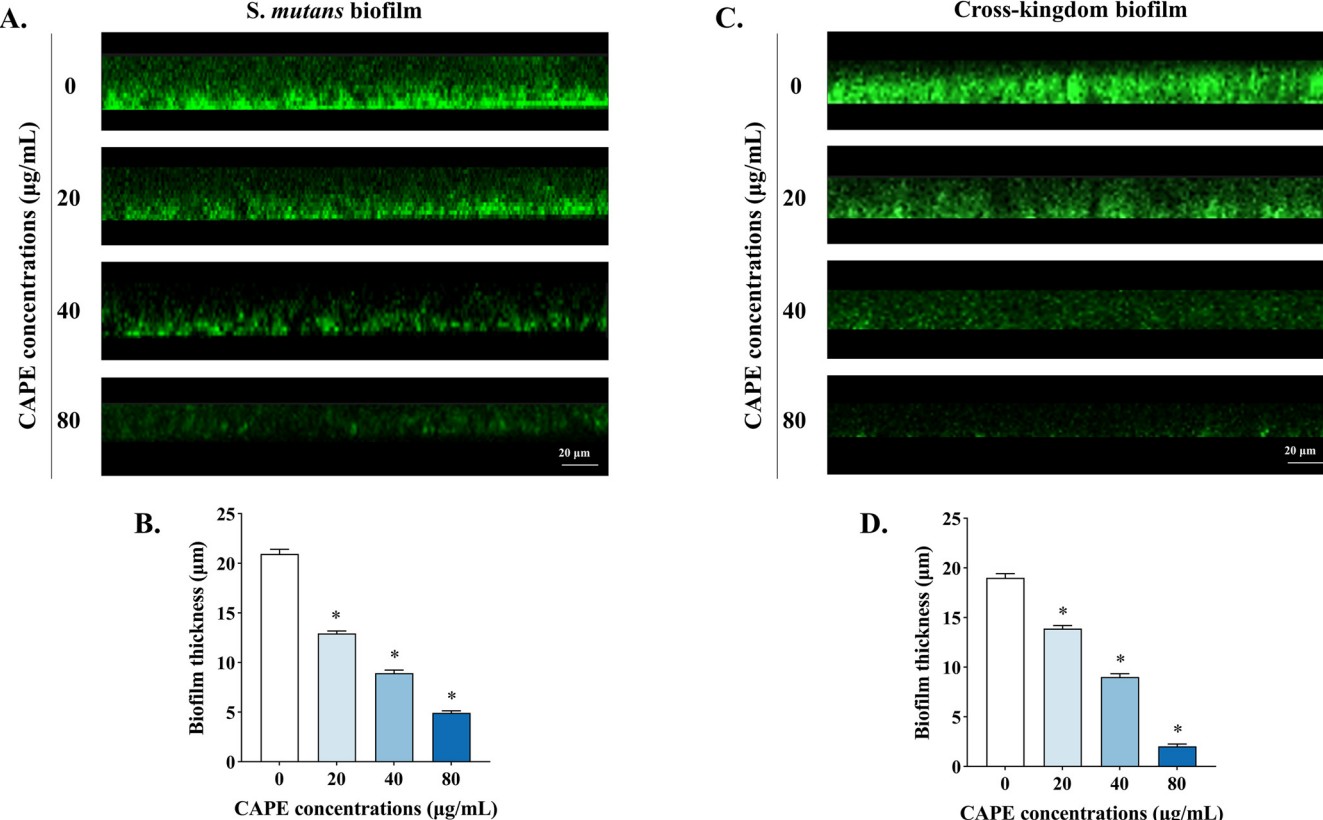

FIG 4 Effect of CAPE on the reduction of *S. mutans* and cross-kingdom biofilms thickness. (A, C) Images of stained biofilms with CAPE treatment obtained by CLSM. The green color (SYTO-9) indicates live biofilms. (B, D) Evaluation of biofilm thickness of *S. mutans* and cross-kingdom biofilms with CAPE penetration. Statistical data are expressed as the mean ± standard deviation (SD) from at least three independent experiments (*, $P < 0.05$).

40 $\mu$g/mL for the same reason. Specifically, CAPE could destroy and lyse biofilms to eliminate dead bacteria.

The thickness of *S. mutans* and cross-kingdom biofilms affected by CAPE is displayed in Fig. 4. The three-dimensional reconstruction of the biofilm architecture indicated that the thickness reduced with the increase in CAPE concentration. The biofilms generated in the control group were approximately 20 $\mu$m on glass coverslips without antimicrobial treatment. With the penetration of CAPE, the biofilm integrity was destroyed, so that both *S. mutans* and cross-kingdom biofilms significantly became thinner. Furthermore, as *S. mutans* and *C. albicans* were inactived and eliminated, the penetration depth of CAPE into biofilms deepened. Generally speaking, the effectiveness of CAPE is closely related to the inhibition of microbial growth and the penetration of biofilms.

**CAPE suppresses EPS synthesis in single-species and cross-kingdom biofilms.** CLSM photographs of the cross-kingdom biofilms formed on glass coverslips with EPS synthesis are shown in Fig. 5A. The dyes SYTO-9 and Alexa Fluor 647 stained the live bacteria green and the EPS red, respectively. The images were processed as three-dimensional reconstructions. In the absence of CAPE, the aggregates of species in the biofilms exhibited high density and compactness, while the aggregates gradually became sparse as the concentration of CAPE increased. The EPS surrounding the species also became less abundant with CAPE treatment. Furthermore, the aggregates and EPS distributed in every layer of the biofilms were analyzed, as shown in Fig. 5B. The whole biomass of bacteria (Fig. 5C) and EPS (Fig. 5D) in the cross-kingdom biofilms was determined for statistical analysis. All of the images and results illustrated that CAPE had inhibitory effects on bacterial proliferation and EPS production, causing decreased accumulation of biofilms.

The amount of water-insoluble EPS production in biofilms detected by the anthrone-

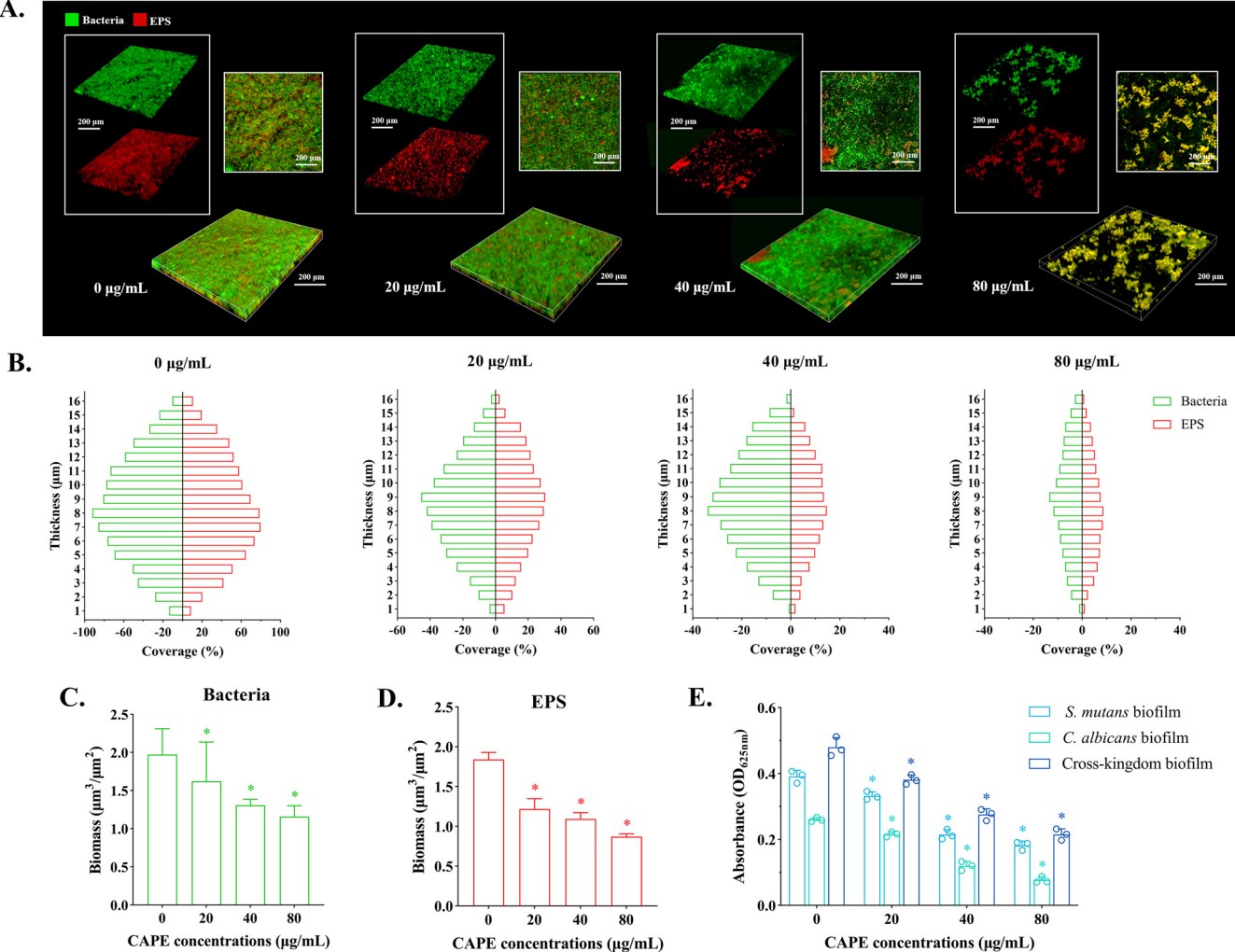

**FIG 5** EPS synthesis in biofilms under CAPE treatment. (A) Images of the cross-kingdom biofilms constructed on glass coverslips with double labeling via confocal laser scanning microscopy. The green color (SYTO-9) and the red color (Alexa Fluor 647) indicate bacteria and EPS in biofilms, respectively. (B) Quantitative distribution of bacteria and EPS in each scanned layer of cross-kingdom biofilms. (C, D) Total biomass of EPS and bacteria in cross-kingdom biofilms of each group. (E) Amount of EPS synthesis in biofilms. Statistical data are expressed as the mean $\pm$ standard deviation (SD) for at least three independent experiments (*, $P < 0.05$).

sulfuric acid assay is presented in Fig. 5E. The absorbance at 625 nm was markedly decreased at all CAPE concentrations compared with the control, which revealed concentration-dependent effects. This illustrated that CAPE with an increasing concentration gradient had an increasing inhibitory effect influence on EPS synthesis in *C. albicans*, *S. mutans* and dual-species biofilms. In addition, in terms of EPS synthesis, the dual-species biofilm demonstrated a cooperative effect in all of the tested groups compared with each single-species biofilm.

**CAPE inhibits EPS synthesis by suppressing *S. mutans* *gtf* gene expression.** Gtfs, as enzymes that ferment sugar, catalyzing the transformation of glucosyl groups, contribute to the synthesis of EPS by *S. mutans*. The expression level of *gtf* genes is closely related to EPS synthesis. The results of the qRT–PCR assay showed the relative expression levels of the *gtf* genes after CAPE treatment (Fig. 6). For *S. mutans* biofilms, CAPE significantly downregulated the gene expression levels of *gtfB*, *gtfC*, and *gtfD* (Fig. 6A). Similarly, the relative gene expression of *gtfB*, *gtfC* and *gtfD* also decreased with increasing CAPE concentration despite the influence of *C. albicans* in the cross-kingdom biofilm (Fig. 6B). Generally, the statistical results revealed the inhibitory effect of CAPE on the expression of *gtf* genes (*gtfB*, *gtfC*, and *gtfD*) in *S. mutans*.

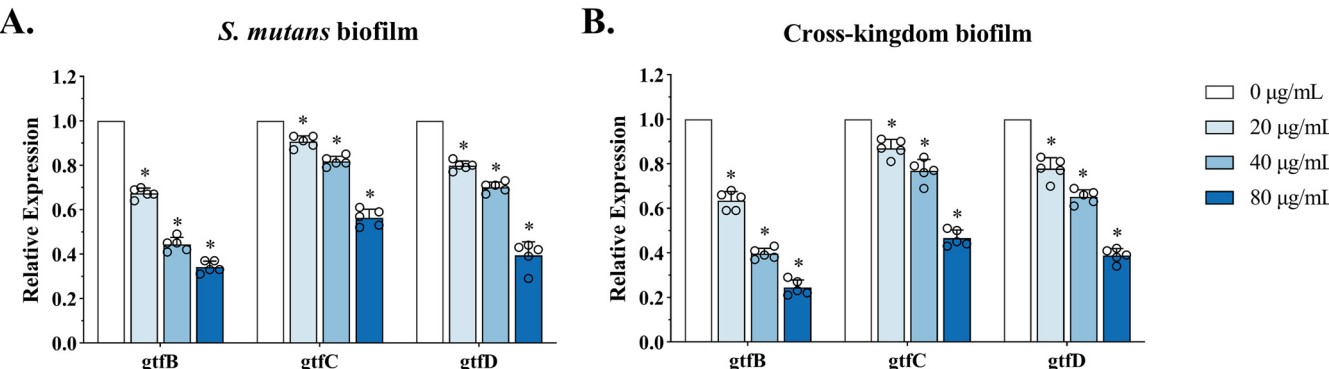

**FIG 6** Inhibitory effects of CAPE on the relative expression of virulence genes in *S. mutans*. Expression of *S. mutans* EPS-related genes (*gtfB*, *gtfC*, and *gtfD*) in *S. mutans* (A) and cross-kingdom (B) biofilms. The relative expression levels of genes were measured by qRT–PCR, and *S. mutans* UA159 16S rRNA was used as the control. Statistical data are presented as the mean $\pm$ standard deviation (SD) from at least three independent experiments (*, $P < 0.05$).

**The biocompatibility of CAPE.** As shown in Fig. 7, the results of the CCK-8 assay demonstrated the biocompatibility of CAPE in the HOK cell line. The results showed a slight decrease in cell viability in the CAPE-treated groups compared with the control group. The 40 $\mu$g/mL and 80 $\mu$g/mL CAPE groups exhibited no significant difference in cell viability. In general, the proliferation of HOK cells could be slightly inhibited with the tested concentrations of CAPE. The viability of HOK cells was above 80% with 40 $\mu$g/mL and 80 $\mu$g/mL CAPE and above 90% on average with a CAPE concentration of 20 $\mu$g/mL. Compared with CHX, which exhibited obvious cytotoxicity (see the Discussion section), the biocompatibility of CAPE is acceptable.

## DISCUSSION

In this study, we identified the antibacterial and antifungal effects of CAPE on *S. mutans* and *C. albicans*, respectively. The concentrations selected depended on the susceptibility assay results. The results demonstrated that CAPE significantly inhibited both of these strains at MICs ranging from 80 to 160 $\mu$g/mL, which is consistent with a previous report (5). Because of the hydrophobicity of the tested CAPE, high concentrations might affect the absorbance, so the experimental results were further verified. Therefore, gradient concentrations of CAPE were applied to explore the influence on biofilm formation, EPS production, and cariogenic gene expression.

Considering the important role of dental caries as a global public health problem

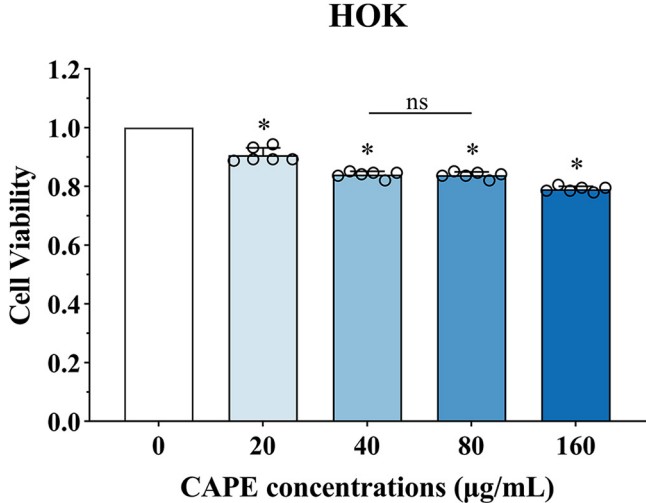

**FIG 7** Viability of HOK cells affected by CAPE. The results of the CCK-8 assay showed the biocompatibility of CAPE at varying concentrations. The statistical data are presented as the means $\pm$ standard deviations (SD) from at least three independent experiments (*, $P < 0.05$; ns, not significant).

(1), more effective control targeting its pathogenesis has long been a research hot spot. In recent years, given the synergistic pathogenicity of multimicrobial communities (2, 25), it has been confirmed that cross-kingdom biofilms of *S. mutans* and *C. albicans* play an essential biological role in severe infection of dental caries (8). In other words, cross-kingdom biofilms are more likely to accelerate the severe demineralization of dental hard tissue, especially in severe ECC (7, 8). There is increasing evidence that bacterial-fungal interactions are key contributors to the cariogenic virulence and drug resistance of biofilms (7, 11, 26). However, current traditional approaches toward biofilm-associated infections still focus on individual bacterial or fungal components, thus limiting their efficacy (13). Similarly, although CAPE has been applied to interfere with caries infection caused by single pathogenic bacteria (5), the use of this natural product remains to be explored for preventing cross-kingdom biofilms with synergistic interactions of *S. mutans* and *C. albicans*.

Therefore, we constructed a cross-kingdom biofilm model of *C. albicans* and *S. mutans* *in vitro*. Inspired by previous studies on the pathogenesis of caries (27), the effects of CAPE on biofilm formation, synthesis of water-insoluble EPS and the expression of related genes have been investigated and analyzed qualitatively and quantitatively. Our study shows that CAPE inhibits biofilm production and EPS synthesis, thereby reducing the biomass of biofilms. Despite CAPE treatment, the presence of *C. albicans* results in an increased number of *S. mutans* colonies in the cross-kingdom biofilm. Our statistical data further support the synergistic promotion effects of *S. mutans* and *C. albicans*, which is consistent with current research (26, 28). Similarly, the increased synthesis of EPS provides more binding sites and stronger adhesion for *S. mutans*, which in turn aggregates with *C. albicans* cells and contributes to increased biomass in biofilms (11, 29). The results also provide convincing evidence that CAPE impedes the secretion of EPS and the adherence of microbial communities in biofilms. Consequently, we showed the potential of CAPE to significantly reduce the virulence of the cross-kingdom biofilm.

The effect of CAPE on coaggregation with *S. mutans* and *C. albicans* was further evaluated through observation and analysis of biofilm morphology. *C. albicans* cells existed in the cross-kingdom biofilm in hyphal form, being enveloped by the extracellular matrix of the *S. mutans* biofilm. The hyphal forms are also regarded as a key virulence factor that promotes the production of *C. albicans* biofilms (24). We found that CAPE not only destroyed the tightness of the biofilms but also deformed the hyphal forms, which in turn led to the reduction in biofilm biomass and structural disruption in cross-kingdom biofilms. Studies have shown that the adhesion between *S. mutans* and *C. albicans* depends on the production of extracellular polymers, especially water-insoluble EPS, contributing to the construction of the extracellular matrix in cariogenic biofilms (8, 30). It is difficult to remove the microbial cells in the mature biofilms encased in the extracellular matrix, as described in a previous report (5). Furthermore, we inspected the influence of CAPE on established mature biofilms with fluorescence staining. The amount of dead bacteria showed an obvious increase in biofilms treated with CAPE. The effect of CAPE in reducing the thickness of biofilms was concentration dependent. Our results revealed that CAPE could devastate and crack the structure of biofilms through penetration of the extracellular matrix. Therefore, the data suggest that CAPE presents a promising advantage for the removal of cross-kingdom biofilms.

Because of the activity of Gtf, EPS form and adhere to biofilms abundantly in cariogenic bacteria. Based on the outcomes of the qRT–PCR assay, we speculate that the inhibitory effect of CAPE on the biofilm and EPS is probably achieved by reducing the expression level of *gtf* genes, including *gtfB*, *gtfC*, and *gtfD*. Notably, the expression of *gtfB* contributes to the adhesion of *S. mutans* and *C. albicans* (11, 26, 31). As shown in other studies *in vitro*, *C. albicans* increased the formation of the extracellular matrix of *S. mutans* by upregulating the expression levels of *gtfB* and *gtfC* in the cross-kingdom biofilm with symbiosis of these two species (26, 32). Therefore, CAPE downregulated the relative expression of the *gtf* gene, providing a theoretical basis for the application of CAPE to suppress severe dental caries caused by *S. mutans* and *C. albicans*.

Our target drug, CAPE, is a novel natural product derived from propolis. Natural products have been a prominent source for discovering new drugs, especially for overcoming the crucial threat posed by antibiotic resistance (18, 33). Some of the natural products discovered have been widely used to prevent and control oral diseases. For example, polyphenols, flavonoids, cinnamaldehyde, cinnamol, and other natural compounds extracted from various plants have been indicated to be effective antibacterial agents against *S. mutans* (17, 34, 35). Moreover, it has been determined that the main bioactive components of propolis can prevent dental caries (5). Studies have indicated that the antimicrobial activity of propolis from different sources could be attributed to the effects of CAPE, the most abundant polyphenolic compound in propolis (5, 36). On this basis, studies have indicated the antimicrobial activity of CAPE. Niu et al. (5) found strong anticariogenic activity of CAPE against pathogenic bacteria, especially *S. mutans*. De Barros et al. (24) discovered that CAPE had antifungal activity against *C. albicans* isolated from oral flora, and similar inhibition of *C. albicans* biofilms was observed *in vitro*. The efficacy of CAPE against oral candidiasis was also detected subsequently in an animal model (24). Our research proved that CAPE exhibited prominent inhibitory activity against the cross-kingdom biofilm formed by *S. mutans* and *C. albicans*, which are considered mutual pathogenic factors.

The biocompatibility of CAPE has been reported by previous studies (21). In our research, CAPE showed slight cytotoxicity to human oral keratinocytes (HOKs), according to the classification (37), which rates extracts as having severe, moderate, or slight cytotoxicity when the viability relative to the control groups is <30%, between 30 and 60%, or >60%, respectively. It is worth noting that CHX is currently considered one of the most common antiseptics in dentistry and the gold standard agent for the application of oral rinses, in particular, the clinical concentration of 0.12% CHX is used to remove plaque biofilm (5, 38, 39). However, its cytotoxicity has been widely reported, especially the obvious reduction of CHX against HOKs and fibroblast cell numbers in a short time (40). Barbara Azzimonti et al. (41) also emphasized that a low concentration of CHX (0.05%) resulted in less than 40% mucosal keratinocytes viability. Thus, our research suggested that the biocompatibility of CAPE is acceptable and has application potential.

The poor water solubility of CAPE has led to a lack of research evidence for drug delivery *in vivo* (42). It also limits the development of CAPE for clinical intervention. Recent studies have proposed the application of nanocarriers for CAPE and other hydrophobic materials to improve bioavailability (43). It has been reported that the application of polymer nanoparticles improves the biocompatibility of propolis, strengthens its antimicrobial activity and allows more efficient drug release *in vivo* (22). In view of the immunoregulatory activity of CAPE, alleviating the inflammatory responses induced by microbial infection (21), more indepth *in vivo* and *in vitro* research is needed, including its combination with biomaterials to improve its application potential.

In conclusion, we found that the natural compound CAPE exhibited significant antimicrobial activity by interfering with the synergistic toxicity of *S. mutans* and *C. albicans*. The specific biomechanisms include inhibitory effects on cross-kingdom biofilm formation and adhesion, EPS synthesis and cariogenic gene expression. The inhibitory effect of CAPE on the cross-kingdom biofilm may contribute to the positive intervention of bacterial-fungal coinfections. CAPE also exhibited acceptable biocompatibility, but more extensive research for clinical safety assessment and efficient application is still warranted. Therefore, CAPE, as a promising natural compound, has the potential to be a new antimicrobial agent to prevent and control dental caries.

## MATERIALS AND METHODS

**CAPE preparation.** CAPE (>97% purity, powder) was purchased from MACKLIN (Shanghai, China). As described in a previous report (5), 50% dimethyl sulfoxide (DMSO, Sigma, Taufkirchen, Germany) was utilized to prepare the CAPE stock solution at a concentration of 12.8 mg/mL.

**Bacterial and fungal strains and growth conditions.** *S. mutans* strain UA159 (ATCC 700610) and *C. albicans* strain SC5314 (ATCC 10691) were acquired from the State Key Laboratory of Oral Diseases, Sichuan University, Chengdu, China. The *C. albicans* and *S. mutans* strains were incubated in YPD medium and brain heart infusion (BHI) medium, respectively, for overnight precultivation at 37°C under

anaerobic conditions with 5% $CO_2$ for recovery (44, 45). The two strains were cocultured in YNBB medium for biofilm formation (28, 29).

**Bacterial and fungal susceptibility assay.** The susceptibility and resistance of microbial strains to specific antibiotics can be described in terms of the MIC (46). The MIC of CAPE for *S. mutans* and *C. albicans* was tested in a 96-well cell culture plate with a modified microdilution method (5). The CAPE stock solution, prepared with 50% DMSO, was added to the corresponding medium with the overnight diluent suspensions ($2 \times 10^6$ CFU/mL for *S. mutans* and $2 \times 10^4$ CFU/mL for *C. albicans*). Therefore, each well finally contained CAPE at concentrations ranging from 10 $\mu$g/mL to 1,280 $\mu$g/mL in 5% DMSO. In addition, 5% DMSO and the corresponding medium alone were used as the solvent and blank controls, respectively. As the positive control, CHX (Sigma–Aldrich, Steinheim, Germany) replaced CAPE in microcells at concentrations ranging from $1.0 \times 10^3$ to $128.0 \times 10^3$ mg/mL according to reported protocols (5). The microplates were cultivated anaerobically with 5% $CO_2$ at 37°C for 46–48 h. The optical density of each microwell at 600 nm (OD600) was estimated in a microplate spectrophotometer (Powerwave XS2, Bio-Tek, USA). Considering the turbidity of high-concentration solutions of CAPE, the plate coating method was applied to subsequently complement operations. Thus, the determined MIC was defined as the lowest concentration of CAPE in the system whose OD600 values did not differ from those of the blank medium group (47).

**CFU counting for planktonic *S. mutans* and *C. albicans* influenced by CAPE.** The CFU counting assay was utilized to quantify the inhibitory effect of CAPE on the planktonic growth of *C. albicans* and *S. mutans* strains by counting the number of live microbial colonies on agar medium plates. In brief, overnight diluent suspensions of each strain were added to a 96-well cell culture plate with the corresponding medium and CAPE. The final concentration of CAPE was adjusted to 20, 40, 80, or 160 $\mu$g/mL. The control group was treated with the solvent rather than CAPE. After incubation for 24 h, the mixed suspensions were diluted $1:10^6$ (for counting *S. mutans*) and $1:10^4$ (for counting *C. albicans*) with sterile phosphate-buffered saline (PBS) in accordance with the reported study (29). Then, *C. albicans* and *S. mutans* were incubated on YPD agar medium plates and BHI agar medium plates, respectively, under anaerobic conditions with 5% $CO_2$ at 37°C for 48 h. The colonies formed on the plates were then counted.

**Growth curves.** Based on the measurement of absorbance, growth curve patterns are generally used for examining the growth and proliferation of microbes over time. This assay was applied to illustrate the short-term inhibitory impact of CAPE against *S. mutans* and *C. albicans* (23, 48). The growth curves of planktonic *C. albicans* and *S. mutans* treated with CAPE were plotted as previously reported (17, 49). In brief, the concentrations of the overnight diluent suspensions of each strain were adjusted to $10^5$ CFU/mL for *S. mutans* in BHI medium and $10^3$ CFU/mL for *C. albicans* in YPD medium, and these were added into a 96-well cell culture plate with CAPE. The final concentrations of CAPE were set at 20 and 40 $\mu$g/mL. The control group was treated with the solvent. The plates were subsequently incubated under anaerobic conditions with 5% $CO_2$ at 37°C for 12 h, and then, the OD600 of each microwell was measured every 30 min using a microplate spectrophotometer.

**Crystal violet staining for biofilm biomass determination.** The crystal violet staining assay was adopted to analyze the effect of CAPE on *C. albicans*, *S. mutans* and cross-kingdom biofilm formation as previously described (50). In brief, the single-species and cross-kingdom biofilms were cultured in 96-well plates as previously described (5) with CAPE at a final concentration of 20, 40, 80, or 160 $\mu$g/mL. The control group was treated with the solvent as described above. After incubation for 24 h, the excess medium in the plates was removed, and every well was washed gently using sterile PBS. After fixation with 100% methanol for 15 min, the adherent biofilms were stained with 0.1% (wt/vol) crystal violet-glacial acetic acid solution for 8 min. After discarding the excess solution and washing three times with PBS again, 95% (vol/vol) ethanol was utilized to extract the crystal violet. Finally, a microplate spectrophotometer was employed to measure the absorbance at 575 nm of each extract in a new 96-well plate.

**CFU counting for quantification of biofilm biomass affected by CAPE.** The CFU counting assay was utilized to quantify the impact of CAPE on the production and formation of *C. albicans*, *S. mutans*, and cross-kingdom biofilms by counting the colonies of live microbes on agar medium plates. In brief, single-species and cross-kingdom biofilms were formed as described above with CAPE at a final concentration of 20 $\mu$g/mL, 40 $\mu$g/mL or 80 $\mu$g/mL. The control group was treated with the solvent. After cultivation in 24-well plates for 24 h, the biofilms were carefully washed with sterile PBS to eliminate planktonic cells. The adherent biofilms were then removed from the floor of each microwell and thoroughly mixed by vortexing with 200 $\mu$L of sterile PBS. After resuspension, each sample was diluted $10^6$-fold to count *S. mutans* colonies and $10^4$-fold to count *C. albicans* colonies. Colonies were counted following incubation on BHI agar medium for *S. mutans* and on YPD agar medium for *C. albicans* (17, 29, 51).

**SEM for biofilm morphology analysis.** Scanning electron microscopy (SEM) was employed to investigate the surface morphology of *C. albicans*, *S. mutans* and cross-kingdom biofilms treated with CAPE. Briefly, glass coverslips were placed into 24-well plates with suspensions of *S. mutans*, *C. albicans*, and both species. These strains were incubated in YNBB medium to final concentrations of $1.0 \times 10^6$ CFU/mL and $1.0 \times 10^4$ CFU/mL, and CAPE was added at a concentration of 20 or 40 $\mu$g/mL. A control group was also prepared. Following incubation for 24 h, the supernatants was discarded, and the biofilms were fixed with 2.5% (vol/vol) glutaraldehyde at 4°C for 10 h, followed by gentle resuspension with PBS. Then, the biofilms were dehydrated in a series of ethanol concentrations (30%, 40%, 50%, 60%, 70%, 80%, 90%, 100%, vol/vol) every 15 min. Finally, the coverslips were dried. After gold spraying, the samples were examined with a scanning electron microscope (Inspect F, FEI, Netherlands) (5, 17). Three points were selected at random in every sample for investigation at three magnifications ($\times 1,000$, $\times 5,000$, $\times 20,000$).

**TABLE 1** Sequences of the primers for qPCR

| Primers | Sequences |
| --- | --- |
| 16S rRNA | Forward: 5'-CCATGTGTAGCGGTGAAATGC-3' |
| | Reverse: 5'-TCATCGTTTACGGCGTGGAC-3' |
| gtfB | Forward: 5'-AGCCGAAAGTTGGTATCGTCC-3' |
| | Reverse: 5'-TGACGCTGTGTTTCTTGGCTC-3' |
| gtfC | Forward: 5'-TTCCGTCCCTTATTGATGACATG-3' |
| | Reverse: 5'-AATTGAAGCGGACTGGTTGCT-3' |
| gtfD | Forward: 5'-TTGACGGTGTTCGTGTTGAT-3' |
| | Reverse: 5'-AAAGCGATAGGCGCAGTTTA-3' |

**Live/dead bacterial viability assay for microscopy and quantitative assays of biofilms.** The live/dead bacterial viability assay was applied to conveniently examine the microstructure of biofilms and monitor the viability of bacterial colonies by measuring the maintenance of membrane integrity. The molecular probes SYTO 9 and propidium iodide (Invitrogen, USA) were adopted to label live bacteria and dead bacteria, respectively. Both dyes were diluted 50-fold with 0.3% (vol/vol) DMSO and then evenly mixed for further use. Specifically, *S. mutans* and cross-kingdom biofilms were formed on glass coverslips in 24-well plates with a CAPE concentration of 20 , 40 , or 80 $\mu$g/mL, and the control group was prepared as described above. The adherent biofilms were washed with PBS after discarding the excess medium and planktonic cells. After drying, the treated coverslips were stained with the mixed dyes following the manufacturer's instructions (Invitrogen). Confocal laser scanning microscopy (CLSM; Olympus FV3000, Japan) was then employed to detect the biofilms. Double-channel scanning was performed with the green channel (at an excitation wavelength of 488 nm) and the red channel (at an excitation wavelength of 561 nm) (52). Three points were randomly selected for the inspection of every specimen, the quantified statistical results of fluorescence intensity and the biofilm thickness of the three-dimensional reconstruction images were assessed using ImageJ COMSTAT software (53, 54).

**Anthrone-sulfuric acid assay for quantitative estimation of EPS.** The anthrone-sulfuric acid assay was applied to verify the inhibitory effect of CAPE treatment on the synthesis of the water-insoluble EPS by *S. mutans*, *C. albicans*, and both species in accordance with reported protocols (5). *S. mutans*, *C. albicans*, and the cross-kingdom biofilms were constructed on the bottom of 24-well plates with a final CAPE concentration of 20, 40, or 80 $\mu$g/mL, and a control group was also prepared. The adherent biofilms were washed carefully using PBS after removing excess solutions, and each was completely collected with 1 mL of PBS in a 1.5 mL centrifuge tube (Corning). Then, the precipitate was obtained by centrifuging at 6,000 $\times$ *g* and 4°C for 10 min. After washing again, the precipitate was resuspended in 0.4 M NaOH with constant agitation at 37°C for 2 h. Then, 3 volumes of anthrone-sulfuric acid reagent was added and further mixed with the supernatant, followed by heating in a water bath at 95°C for 6 min until the reaction was completed. Finally, equal volumes of each sample were delivered to 96-well plates to measure the absorbance at 625 nm with a microplate spectrophotometer.

**CLSM for EPS and bacteria in biofilms.** CLSM was utilized to conveniently observe biofilm formation and quantify the relationship between bacteria and water-insoluble EPS in biofilms. The molecular probes Alexa Fluor 647 and SYTO 9 (Invitrogen, USA) were applied to mark the EPS and live bacteria, respectively. First, *S. mutans* and cross-kingdom biofilms were formed with the same method and simultaneously mixed with the fluorochrome Alexa Fluor 647. Then, adherent biofilms were washed carefully with PBS without the excess solutions and stained with the fluorochrome SYTO 9 for 5 min. After washing again, the coverslips were observed using CLSM (Olympus FV3000, Japan). Double-channel scanning inspections were performed with the green channel (at an excitation wavelength of 488 nm) and the red channel (at an excitation wavelength of 561 nm). Three random points were selected for the investigation of each sample, and the quantified fluorescence intensity was assessed with ImageJ COMSTAT software (53).

**qRT–PCR analysis of cariogenic gene expression in biofilms.** The relative expression levels of cariogenic virulence genes, including *gtfB*, *gtfC* and *gtfD*, were analyzed with a c assay in *S. mutans* and cross-kingdom biofilms under CAPE treatment. In brief, *S. mutans* and cross-kingdom biofilms were cultured in YNBB medium with CAPE at a final concentration of 20, 40, or 80 $\mu$g/mL, with sterile PBS serving as the control group, for 24 h. The adherent biofilms were collected using the same experimental method. The total RNA of *S. mutans* UA159 was extracted with TRIzol reagent (Invitrogen, USA). After that, RNA reverse transcription was performed with the PrimeScript RT reagent kit (TaKaRa, Japan). The relative expression levels of *gtfB*, *gtfC* and *gtfD* mRNA were determined using the qRT–PCR assay, with *S. mutans* UA159 16S rRNA used as the reference. The primers were synthesized by TsingKe Biotechnology, and the sequences are presented in Table 1. In general, the quantitative PCR conditions and procedures were similar to those used in a previous study (17). The $2^{-\Delta\Delta CT}$ method was utilized to calculate the relative mRNA expression levels of *S. mutans gtf* genes (55). The assay was independently performed in triplicate according to the qRT–PCR procedures and experimental conditions.

**CCK-8 assay for biocompatibility.** The biocompatibility of CAPE on human oral keratinocytes (HOKs) was detected by the Cell-Counting-Kit 8 (CCK-8) assay according to the manufacturer's instructions (Beijing Solarbio Science & Technology Co., Ltd.). The HOK cell line was acquired from the State Key Laboratory of Oral Diseases, Sichuan University, Chengdu, China. The cells were incubated in 100 U/mL penicillin

supplemented with DMEM (Gibco) and 100 mg/mL streptomycin supplemented with 20% FBS (Gibco) in a volume of 100 $\mu$L per well in a 96-well cell plate with a final CAPE concentration of 20, 40, 80, or 160 $\mu$g/mL, the control group was prepared as described above. After incubation for 24 h, the cells were resuspended gently with sterile PBS, and 10 $\mu$L of CCK-8 reagent was mixed in every well. After another 4 h of cultivation, the absorbance at 450 nm was measured using a microplate spectrophotometer.

**Statistical analysis.** All experiments were carried out independently three times, with reliable data obtained from repeat wells for each concentration. The statistical analysis was performed using SPSS software (IBM SPSS Statistics 25, USA). The statistical results are presented as the mean $\pm$ standard deviation (SD). One-way analysis of variance and a *post hoc t* test were applied to analyze the difference between the experimental groups and the control group. A value of $P < 0.05$ was considered to be statistically significant.

## ACKNOWLEDGMENTS

This work was supported by Research Funding for Talent Development, West China Hospital of Stomatology, Sichuan University (RCDWJS2021-6) and Research and Development Funding, West China Hospital of Stomatology, Sichuan University (RD-02-202009).

We declare that there are no conflicts of interest.

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
