## [Reviewer comments · Microbiology Spectrum]

Microbiology Spectrum

Caffeic Acid Phenethyl Ester (CAPE) Inhibits Cross-Kingdom Biofilm Formation of *Streptococcus mutans* and *Candida albicans*

Wumeng Yin, Zhong Zhang, Xinxing Shuai, Xuedong Zhou, and Derong Yin

Corresponding Author(s): Derong Yin, State Key Laboratory of Oral Diseases, National Clinical Research Center for Oral Diseases, West China Hospital of Stomatology, Sichuan University

Review Timeline:

Submission Date:	April 29, 2022
Editorial Decision:	May 27, 2022
Revision Received:	July 20, 2022
Accepted:	August 2, 2022

Editor: Xiaoyu Tang

Reviewer(s): Disclosure of reviewer identity is with reference to reviewer comments included in decision letter(s). The following individuals involved in review of your submission have agreed to reveal their identity: Jonathon L Baker (Reviewer #3)

Transaction Report:

DOI: <https://doi.org/10.1128/spectrum.01578-22>

May 27, 2022

Dr. Wumeng Yin
Sichuan University
West China Hospital of Stomatology
Section 3 RenMin South Road
chengdu
China

Re: Spectrum01578-22 (Caffeic Acid Phenethyl Ester (CAPE) Inhibits Cross-Kingdom Biofilm Formation of Streptococcus mutans and Candida albicans)

Dear Dr. Wumeng Yin:

Link Not Available

Sincerely,

Xiaoyu Tang

Journals Department
Reviewer comments:

Reviewer #2 (Comments for the Author):

Reviewer comment for Spectrum01578-22 (Caffeic Acid Phenethyl Ester (CAPE) Inhibits Cross-Kingdom Biofilm Formation of Streptococcus mutants and Candida Albicans)

It is well known that Candida albicans and Streptococcus mutans interact to form robust biofilm, which in turn degrade enamel to cause cavities. Therefore, determining mechanisms to disrupt and these biofilms is crucial in regulating their disease outcome. The current manuscript employed Caffeic Acid Phenethyl Ester (CAPE), a naturally existing compound, to disrupt and kill both

bacteria and their cross-kingdom biofilm formation. Authors carefully monitored planktonic and biofilm growth and EPS formation in presence of increasing concentration of CAPE. Although some of the explanation and argument of the data can be improved (see below), this article can help progressing the current field.

Comments

1. Authors argue that CAPE suppressed growth, biofilm formation and EPS synthesis via interactions with the two species. However, in their experiments, they did not show direct interaction of the CAPE with the two species. This has to be shown by actual experiment.
2. How using CAPE against biofilm formation will affect its other activities in eukaryotic organisms? Most importantly, CAPE has immunomodulatory activity, which may impact the eukaryotic host physiology unintended way. In addition to testing cytotoxicity, how immune activity may go a long way for using this product as therapeutic.
3. CAPE has effect on cross-kingdom biofilm formation of *S. mut* and *C. albicans* in vitro experiments. How about in vivo biofilms? How this cross-kingdom biofilm is different in in vivo and whether CAPE can still function in the native environment?
4. In Figure 1, panel A and B show that *C. albicans* grow in 40 µg/mL CAPE despite in low numbers. However, in Figure 1D, OD600 show that under the same concentration, no growth can be observed. What is the reason?
5. In Figure 4, can the decrease in EPS formation be due to not CAPE inhibiting EPS formation but CAPE killing the bacteria? Both biomass and EPS amount decreased simultaneously. In other words, can decrease in EPS be due to lower number of bacteria? How can the authors argue one versus the other? The given experiment is not enough to distinguish between these two possibilities. Perhaps, the authors need to normalize EPS amount to per bacterial cell bases to argue this point?
6. What is the CAPE biofilm penetration depth? Can you label CAPE and measure how deep it can penetrate the biofilm?
7. Authors argue that HOK cell viability was minimally affected by CAPE, but Figure 6 shows statistically significant difference in all concentrations tested when compared to control (no CAPE experiment). This is counterintuitive and show that CAPE are toxic to eukaryotic cells. Also, at 40 and 80 µg/mL (MIC), the HOK cells had 80% viability which means 20% of the HOK cells were killed. By therapeutic standards, this is very high and cannot be used in humans.
8. What is the MIC of CAPE for other strains of *S. mut* and *C. albicans*. Are they similar to UA159 and SC5314? How did you pick these strains for the study?
9. How can a CFU be determined from a biofilm in Figure 2B? Don't they clump-up and form aggregates, giving false CFU?
10. In Figure 5 and lines 182-188, authors need to explain what *gtf* genes do? And why they decide to test for the expression level of these genes? Did the authors do comprehensive screen for other cariogenic genes?
11. It will go a long way to show additional oral epithelial cell tests. HOK cells are one of many oral keratinocytes and they may show different effect compare to other cell lines.

Reviewer #3 (Comments for the Author):

In this study, the authors examine the effects of caffeic acid phenethyl ester on single and multispecies cultures and biofilms of the caries pathogens *Streptococcus mutans* and *Candida albicans*. Overall, the rationale and methods used are sound, and most of the results are supported by the conclusions. The study is well-written, and fits nicely within the scope of *Microbiology Spectrum*. There are a couple of issues that do need to be addressed before the article is suitable for publication, as described in the following points:

1. Abstract: ASM Journals not use sectioned abstracts, please combine and reword accordingly.
2. Importance: "closely-related to the cross-kingdom", this sentence has awkward phrasing as currently written, please revise.
3. Line 57: please change "the key pathogen" to "a key pathogen", as many researchers in the field would consider this an exaggeration of the role of *S. mutans*, as cases of caries without detectable levels of *S. mutans* are routine.
4. Line 110: "corresponding colony variations" please clarify or change? Do you mean number of CFUs or differences in the actual colony morphology?
5. Line 156: "was affected by CAPE" awkward phrasing again here...I think you mean something more like "a reduction in biofilm accumulation correlated with increasing concentration of CAPE"
6. qRT-PCR: This is important! If I understand the methods correctly, the relative expression of *gtfB*, *C*, and *D* in each sample is "relative" to the corresponding 0µg CAPE sample. Since there are clearly fewer cells, and more dead cells in the higher CAPE concentration samples, you would expect less *gtf* RNA, and indeed less RNA overall in those samples. Crucially, using this method, you therefore cannot hypothesize that CAPE specifically reduced the transcription of *gtfs*. You would need to control for the differences in cell viability and/or also compare to the RNA levels of other genes. Please either add experiments with the proper controls or remove this experiment from the study.
7. Biocompatibility: It is not clear from the current manuscript what current "acceptable" levels of CCK-8 viability are, and how

much toxicity in the assay might translate to toxicity in vivo. Please either remove claims of "good biocompatibility" or add references explaining what "good biocompatibility" is and how the results here compare to it.

8. Line 218: please remove "promising", I think that's a bit overboard at this stage.

9. Lines 251-257: see comment #6 above

10. Line 265: do you mean "plants" instead of "implants"?

11. Line 277: see comment #7 above.

12. Lines 371-372: Since I don't believe that YPD and BHI are selective, how do you differentiate between colonies of *S. mutans* and *C. albicans* by plating on solid media from the co-culture? This is unclear, please explain.

Staff Comments:

Preparing Revision Guidelines

Please return the manuscript within 60 days; if you cannot complete the modification within this time period, please contact me. If you do not wish to modify the manuscript and prefer to submit it to another journal, please notify me of your decision immediately so that the manuscript may be formally withdrawn from consideration by Microbiology Spectrum.

Reviewer comment for Spectrum01578-22 (Caffeic Acid Phenethyl Ester (CAPE) Inhibits Cross-Kingdom Biofilm Formation of Streptococcus mutants and Candida Albicans)

It is well known that *Candida albicans* and *Streptococcus mutans* interact to form robust biofilm, which in turn degrade enamel to cause cavities. Therefore, determining mechanisms to disrupt and these biofilms is crucial in regulating their disease outcome. The current manuscript employed Caffeic Acid Phenethyl Ester (CAPE), a naturally existing compound, to disrupt and kill both bacteria and their cross-kingdom biofilm formation. Authors carefully monitored planktonic and biofilm growth and EPS formation in presence of increasing concentration of CAPE. Although some of the explanation and argument of the data can be improved (see below), this article can help progressing the current field.

Comments

1. Authors argue that CAPE suppressed growth, biofilm formation and EPS synthesis via interactions with the two species. However, in their experiments, they did not show direct interaction of the CAPE with the two species. This has to be shown by actual experiment.
2. How using CAPE against biofilm formation will affect its other activities in eukaryotic organisms? Most importantly, CAPE has immunomodulatory activity, which may impact the eukaryotic host physiology unintended way. In addition to testing cytotoxicity, how immune activity may go a long way for using this product as therapeutic.
3. CAPE has effect on cross-kingdom biofilm formation of *S.mut* and *C. albicans* in vitro experiments. How about in vivo biofilms? How this cross-kingdom biofilm is different in in vivo and whether CAPE can still function in the native environment?
4. In Figure 1, panel A and B show that *C. albicans* grow in 40 ug/mL CAPE despite in low numbers. However, in Figure 1D, OD600 show that under the same concentration, no growth can be observed. What is the reason?
5. In Figure 4, can the decrease in EPS formation be due to not CAPE inhibiting EPS formation but CAPE killing the bacteria? Both biomass and EPS amount decreased simultaneously. In other words, can decrease in EPS be due to lower number of bacteria? How can the authors argue one versus the other? The given experiment is not enough to distinguish between these two possibilities. Perhaps, the authors need to normalize EPS amount to per bacterial cell bases to argue this point?
6. What is the CAPE biofilm penetration depth? Can you label CAPE and measure how deep it can penetrate the biofilm?
7. Authors argue that HOK cell viability was minimally affected by CAPE, but Figure 6 shows statistically significant difference in all concentrations tested when compared to control (no CAPE experiment). This is counterintuitive and show that CAPE are toxic to eukaryotic cells. Also, at 40 and 80 ug/mL (MIC), the HOK cells had 80% viability which means 20% of the HOK cells were killed. By therapeutic standards, this is very high and cannot be used in humans.
8. What is the MIC of CAPE for other strains of *S.mut* and *C.albicans*. Are they similar to UA159 and SC5314? How did you pick these strains for the study?
9. How can a CFU be determined from a biofilm in Figure 2B? Don't they clump-up and form aggregates, giving false CFU?
10. In Figure 5 and lines 182-188, authors need to explain what *gtf* genes do? And why they decide to test for the expression level of these genes? Did the authors do comprehensive screen for other cariogenic genes?
11. It will go a long way to show additional oral epithelial cell tests. HOK cells are one of many oral keratinocytes and they may show different effect compare to other cell lines.

Response to Reviewer #2

Dear Reviewer,

We thank you for your professional review of our article. There were several issues that needed to be addressed. According to your helpful suggestions, we have made extensive modifications to our previous manuscript and supplemented additional data to make our results more convincing; the detailed corrections are listed. Please see below, in blue text, our point-by-point responses to your comments and concerns. All page numbers refer to the revised manuscript file with tracked changes.

Comment 1:

Authors argue that CAPE suppressed growth, biofilm formation and EPS synthesis via interactions with the two species. However, in their experiments, they did not show direct interaction of the CAPE with the two species. This has to be shown by actual experiment.

Response 1:

We gratefully appreciate your careful review and noting our inaccurate statement. We apologize for the misunderstanding caused by our inaccurate representation of the role of CAPE and would like to explain it in more detail.

According to reported studies (1-3), the interactions of *S. mutans* and *C. albicans* can enhance cariogenicity, which is consistent with our research basis. Our study displayed the inhibitory effects of CAPE on *S. mutans* and *C. albicans*, which corresponded to previous studies (4, 5). Therefore, we hypothesized that CAPE could also play an inhibitory role in the co-culture systems. Through confirmed experiments, CAPE can indeed reduce the cariogenicity of *S. mutans* and *C. albicans* co-culture systems by suppressing growth, biofilm formation and EPS synthesis.

However, our results were misinterpreted as the interaction of CAPE with the two species. We apologize for the inaccurate text and have made the following two modifications.

The first modification is shown in **Lines 31-33**.

We changed

“As a result, CAPE suppressed the growth, biofilm formation and EPS synthesis of *C. albicans* and *S. mutans* via interactions with the two species.”

to

“The results showed that CAPE suppressed the growth, biofilm formation and extracellular polysaccharides (EPS) synthesis of *C. albicans* and *S. mutans* in the co-culture system of the two species.”

the CCK-8 assay. The results showed that CAPE suppressed the growth, biofilm formation and
extracellular polysaccharides (EPS) synthesis of *C. albicans* and *S. mutans* in the co-culture
system of the two species. The expression of the *gff* gene was also suppressed by CAPE. The

The second modification is shown in **Lines 89-91**.

We changed

“In general, the number of studies on the interaction of CAPE with *C. albicans* and *S. mutans* and the effects of CAPE on the cariogenic virulence of cross-kingdom biofilms is limited, which prompted us to study this topic further.”

to

“In general, the effects of CAPE on the cariogenic virulence of cross-kingdom biofilms formed by *C. albicans* and *S. mutans* are still unclear, which prompted us to study this topic further.”

state of both species. In general, the effects of CAPE on the cariogenic virulence of cross-
kingdom biofilms formed by *C. albicans* and *S. mutans* are still unclear, which prompted us to
study this topic further.

Comment 2:

How using CAPE against biofilm formation will affect its other activities in eukaryotic organisms? Most importantly, CAPE has immunomodulatory activity, which may impact the eukaryotic host physiology unintended way. In addition to testing cytotoxicity, how immune activity may go a long way for using this product as therapeutic.

Response 2:

Thank you for your valuable suggestion. As a type of multifunctional active substance, CAPE has been extensively studied (6, 7), especially with regard to its immunomodulatory activity, as you mentioned. Our research concentrated on the antimicrobial activity of CAPE and its potential application value in the prevention of severe caries, which was consistent with most previous studies on the antibacterial and antifungal effects of CAPE (4, 8-10).

On the basis of the current relevant report (11, 12), antimicrobial agents are usually

used as mouthwash additives or toothpaste additives in daily oral protection. Taking the commonly used hydrogen peroxide (H₂O₂) mouthwash as an example (13), although H₂O₂ has important biological functions as a reactive oxygen species, it is still used as a mouthwash additive.

Therefore, from the perspective of this study, we will consider the further application of relevant research in the future from the dosages, dosage form, dosage regimen and other aspects of CAPE to conduct a more comprehensive evaluation. Accordingly, for further study on the immunomodulatory activity of CAPE, we provide a supplementary exposition in the Discussion of the article (**Line 308-311**).

efficient drug release in vivo (22). In view of the immunoregulatory activity of CAPE,
alleviating the inflammatory responses induced by microbial infection (21), more in-depth in
vivo and in vitro research is needed, including its combination with biomaterials to improve its
application potential.↵

Comment 3:

CAPE has effect on cross-Kingdom biofilm formation of *S. mutans* and *C. albicans* in vitro experiments. How about in vivo biofilms? How this cross-kingdom biofilm is different in in vivo and whether CAPE can still function in the native environment?

Response 3:

We sincerely thank you for this important comment, which we also considered at the beginning of our research design. Therefore, we will explain our consideration in detail as follows.

Although CAPE has significant advantages in biological activities, its low water solubility leads to poor bioavailability, which limits its therapeutic application and in vivo model construction (6, 14). Studies have shown that CAPE can be hydrolysed into caffeic acid by esterase in vivo (15), which observably limits the effectiveness and often requires high doses to achieve the therapeutic effect.

In our research on the antimicrobial activity of CAPE, although we agree that the in vivo biofilm model is an important consideration, it is beyond the scope of this study. The oral cavity is a moist and dynamic microenvironment, which presents in vivo experiments with the potential inefficiency of the simple CAPE dosage form.

Currently, applications of biomaterials and modern drug delivery systems are gradually being developed to address CAPE's poor solubility and bioavailability. For

example, the application of nanocarriers for CAPE could achieve better therapeutic effects at lower doses of CAPE (16). This explanation of the limitations of CAPE and the prospects for future research are mentioned in the Discussion section (Lines 303-311). Furthermore, we are planning to explore the combination of CAPE and nanotechnology in the microenvironment.

The poor water solubility of CAPE has led to a lack of research evidence for drug
delivery in vivo (42). It also limits the development of CAPE for clinical intervention. Recent
studies have proposed the application of nanocarriers for CAPE and other hydrophobic materials
to improve bioavailability (43). It has been reported that the application of polymer nanoparticles
improves the biocompatibility of propolis, strengthens its antimicrobial activity and allows more
efficient drug release in vivo (22). In view of the immunoregulatory activity of CAPE,
alleviating the inflammatory responses induced by microbial infection (21), more in-depth in
vivo and in vitro research is needed, including its combination with biomaterials to improve its
application potential.[↵]

Comment 4:

In Figure 1, panel A and B show that *C. albicans* grow in 40 ug/mL CAPE despite in low numbers. However, in Figure 1D, OD600 show that under the same concentration, no growth can be observed. What is the reason?

Response 4:

Thank you for this question, which I will address as follows.

First, we carefully checked the data in Figure 1 and ensured its authenticity. The plate coating experiment shown in panels A and B is based on the determination of MIC, which is referred to as a more intuitive and accurate supplement to the antimicrobial effect of CAPE at gradient concentrations. Therefore, the plate-coating results tend to reflect the more subtle microbial growth in the system.

The growth of *C. albicans* at 40 µg/mL CAPE shown in panel D presented as a flat curve, similar to the stationary phase in which the bacterial growth reached a plateau as the number of dying cells equalled the number of dividing cells. This finding illustrated that the growth and proliferation of *C. albicans* were initially significantly inhibited by CAPE; thus, the fungal density remained almost constant. However, the sluggish growth of *C. albicans* does not mean that the *C. albicans* in the original system were completely killed by CAPE; there is no obvious recession that would be considered the death phase. Therefore, the CAPE-treated group at 40 µg/mL showed the presence of *C. albicans* in panels A and B.

In view of our unclear text in the Results, we have modified the text as shown in **Lines 115-117**.

We changed

“With regard to the growth of *C. albicans*, the optical density in the system indicated that *C. albicans* hardly grew with 1/2 MIC CAPE.”

to

“With regard to the growth of *C. albicans*, the optical density in the system indicated that *C. albicans* hardly proliferated and grew sluggishly with 1/2 MIC CAPE.”

an approximately 2-fold reduction in the OD600 of *S. mutans*. With regard to the growth of *C.*
*albicans*, the optical density in the system indicated that *C. albicans* hardly proliferated and grew
sluggishly with 1/2 MIC CAPE. The growth and proliferation of both species were suppressed

Comment 5:

In Figure 4, can the decrease in EPS formation be due to not CAPE inhibiting EPS formation but CAPE killing the bacteria? Both biomass and EPS amount decreased simultaneously. In other words, can decrease in EPS be due to lower number of bacteria? How can the authors argue one versus the other? The given experiment is not enough to distinguish between these two possibilities. Perhaps, the authors need to normalize EPS amount to per bacterial cell bases to argue this point?

Response 5:

We appreciate this meaningful question, and we will elaborate on these two valid arguments.

On the one hand, we have directly confirmed that CAPE can kill bacteria in biofilms through the live/dead bacterial viability assay using CLSM. On the other hand, the qRT-PCR assay was used to verify the inhibitory effect of CAPE on EPS synthesis, rather than decreasing the amount of EPS simply by reducing the living bacteria. The results demonstrated that CAPE could significantly downregulate the expression level of *Gtf* genes (*gtfB*, *gtfC*, *gtfD*), which guide the production of EPS through the synthesis of Gtf enzymes. In other words, the outcomes of the qRT-PCR assay indirectly proved that CAPE could inhibit EPS formation by controlling the expression of corresponding cariogenic genes.

Consequently, we believe that CAPE reduces biofilm biomass by simultaneously killing bacteria and inhibiting EPS formation. These methods have also been demonstrated in other studies on the antimicrobial properties of natural products or

extracts (17, 18).

Thank you again, and we sincerely hope our response has resolved this question.

Comment 6:

What is the CAPE biofilm penetration depth? Can you label CAPE and measure how deep it can penetrate the biofilm?

Response 6:

We appreciate your valuable comment.

In our opinion, CAPE biofilm penetration depth is the key to the effectiveness of CAPE because CAPE penetration is directly related to microbial inactivation and the control of biofilms (19). In our research, confocal laser scanning microscopy (CLSM) was applied to observe the stained biofilms, as shown in Figure 3. This procedure only illustrated the changes in the surface structure of biofilms and the quantitative analysis of fluorescence.

Therefore, according to your suggestion, we have made some supplements as follows. Briefly, we used ImageJ COMSTAT software to reconstruct the three-dimensional architecture of biofilms and quantitatively measure the thickness of biofilms to evaluate CAPE biofilm penetration depth, which is consistent with the reported references (9, 20).

Then, we modified the Figures, Materials and Methods, and Results.

In the Figures, we added “**Fig. 4.**” and we adjusted the sequence of other figures from Fig. 5 to Fig. 7.

Fig. 4.

As shown in **Lines 664-668**,

We added the following text: “**Fig. 4. Effect of CAPE on the reduction of *S. mutans* and cross-kingdom biofilms thickness.** (A) (C) Images of stained biofilms with CAPE treatment obtained by CLSM. The green colour (SYTO-9) indicates live biofilms. (B) (D) Evaluation of biofilm thickness of *S. mutans* and cross-kingdom biofilms with CAPE penetration. Statistical data are expressed as the mean ± standard deviation (SD) from at least three independent experiments (*p < 0.05).”

**Fig. 4. Effect of CAPE on the reduction of *S. mutans* and cross-kingdom biofilms thickness.** (A) (C)
 Images of stained biofilms with CAPE treatment obtained by CLSM. The green colour (SYTO-9) indicates
 live biofilms. (B) (D) Evaluation of biofilm thickness of *S. mutans* and cross-kingdom biofilms with CAPE
 penetration. Statistical data are expressed as the mean ± standard deviation (SD) from at least three
 independent experiments (*p < 0.05).⁴

In the Materials and Methods section, as shown in **Lines 425-428**,

we changed

“Three points were randomly selected for the inspection of every specimen, and the quantified statistical results of fluorescence intensity were assessed using ImageJ COMSTAT software.”

to

“Three points were randomly selected for the inspection of every specimen, the quantified statistical results of fluorescence intensity and the biofilm thickness of the three-dimensional reconstruction images were assessed using ImageJ COMSTAT software.”

We also added these important references (9, 20).

- 425 red channel (at an excitation wavelength of 561 nm) (52). Three points were randomly selected
426 for the inspection of every specimen, the quantified statistical results of fluorescence intensity
427 and the biofilm thickness of the three-dimensional reconstruction images were assessed using
428 ImageJ COMSTAT software (53, 54).[↵]
53. Morris AJ, Li A, Jackson L, Yau YCW, Waters V. 2020. Quantifying the Effects of Antimicrobials on In vitro Biofilm Architecture using COMSTAT Software. J Vis Exp. <https://doi.org/10.3791/61759>.[↵]
54. Veloz JJ, Alvear M, Salazar LA. 2019. Antimicrobial and Antibiofilm Activity against *Streptococcus mutans* of Individual and Mixtures of the Main Polyphenolic Compounds Found in Chilean Propolis. Biomed Res Int 2019:7602343.[↵]

In the Results section, we added the corresponding results, as shown in **Lines 162-170 as follows**.

“The thickness of *S. mutans* and cross-kingdom biofilms affected by CAPE is displayed in Fig. 4. The three-dimensional reconstruction of the biofilm architecture indicated that the thickness reduced with the increase in CAPE concentration. The biofilms generated in the control group were approximately 20 µm on glass coverslips without antimicrobial treatment. With the penetration of CAPE, the biofilm integrity was destroyed, so that both *S. mutans* and cross-kingdom biofilms significantly became thinner. Furthermore, as *S. mutans* and *C. albicans* were inactivated and eliminated, the penetration depth of CAPE into biofilms deepened. Generally speaking, the effectiveness of CAPE is closely related to the inhibition of microbial growth and the penetration of biofilms.”

The thickness of *S. mutans* and cross-kingdom biofilms affected by CAPE is displayed in
Fig. 4. The three-dimensional reconstruction of the biofilm architecture indicated that the
thickness reduced with the increase in CAPE concentration. The biofilms generated in the
control group were approximately 20 μm on glass coverslips without antimicrobial treatment.
With the penetration of CAPE, the biofilm integrity was destroyed, so that both *S. mutans* and
cross-kingdom biofilms significantly became thinner. Furthermore, as *S. mutans* and *C. albicans*
were inactivated and eliminated, the penetration depth of CAPE into biofilms deepened. Generally
speaking, the effectiveness of CAPE is closely related to the inhibition of microbial growth and
the penetration of biofilms.⁴

Comment 7:

Authors argue that HOK cell viability was minimally affected by CAPE, but Figure 6 shows statistically significant difference in all concentrations tested when compared to control (no CAPE experiment). This is counterintuitive and show that CAPE are toxic to eukaryotic cells. Also, at 40 and 80 $\mu\text{g}/\text{mL}$ (MIC), the HOK cells had 80% viability which means 20% of the HOK cells were killed. By therapeutic standards, this is very high and cannot be used in humans.

Response 7:

Thank you for your valuable suggestion; our explanation is as follows.

In this study, we chose chlorhexidine (CHX) as the positive control. CHX is considered the gold standard for application in oral rinses and has been extensively used to remove plaque biofilms in clinical dentistry; in particular, a 0.12% concentration of CHX is used to prevent caries occurrence and progression (4, 21, 22). However, its cytotoxicity remains a disadvantage that has not been well resolved (23). According to reported research (24), a low concentration of CHX (0.05%) resulted in less than 40% oral keratinocyte viability.

We carefully verified the data in Figure 6 to ensure its authenticity. At a 20 $\mu\text{g}/\text{mL}$ concentration of CAPE, human oral keratinocytes (HOKs) exhibited approximately 90% viability. Additionally, the HOK cells had more than 80% viability at 40 $\mu\text{g}/\text{mL}$ and 80 $\mu\text{g}/\text{mL}$ (MIC). According to the classification of cytotoxicity (25), extracts are considered severely, moderately or slightly cytotoxic when the viability relative to the controls is less than 30%, between 30% and 60%, or greater than 60%, respectively. Therefore, CAPE illustrated “slight cytotoxicity” in our research instead of good biocompatibility, which was corrected as shown in **Line 292**.

292 research, CAPE showed slight cytotoxicity to human oral keratinocytes (HOKs), according to the

Although all CAPE-treated groups showed a significant difference from the control group, the 80% oral keratinocyte viability with CAPE treatment was significantly higher than that treated with a low concentration of CHX below 40%. Additionally, our supplementary susceptibility assay data show that the MIC of CHX ranged from 4×10^3 to 8×10^3 mg/mL for *S. mutans* and *C. albicans*, which is significantly higher than the MIC concentration of CAPE. Therefore, our research suggested that the biocompatibility of CAPE is acceptable and has application potential.

Accordingly, we have made modifications in the Abstract, Materials and Methods, Results, and Discussion.

In the Abstract, we changed “good” to “acceptable” in **Lines 34-35**.

efficacy of CAPE was concentration dependent, and the compound exhibited acceptable
biocompatibility. Our research lays the foundation for further study of the application of the

In the Materials and Methods, as shown in **Lines 342-344**, we changed

“CHX (Sigma–Aldrich, Steinheim, Germany) replaced CAPE in microcells as the positive control according to reported protocols”

to

“As the positive control, CHX (Sigma–Aldrich, Steinheim, Germany) replaced CAPE in microcells at concentrations ranging from 1.0×10^3 to 128.0×10^3 mg/mL according to reported protocols.”

used as the solvent and blank controls, respectively. As the positive control, CHX (Sigma–
Aldrich, Steinheim, Germany) replaced CAPE in microcells at concentrations ranging from
1.0×10^3 to 128.0×10^3 mg/mL according to reported protocols (5). The microplates were

In the Results section, there are three changes.

The first addition is shown in **Lines 103-105**.

We added “The MIC of CHX ranged from 4×10^3 to 8×10^3 mg/mL for both strains as the positive control, which was significantly higher than the MIC concentration of CAPE.”

control had no effect on bacterial and fungal growth. The MIC of CHX ranged from 4×10^3 to
8×10^3 mg/mL for both strains as the positive control, which was significantly higher than the
MIC concentration of CAPE. Thus, 20, 40, 80 and 160 $\mu\text{g/mL}$ CAPE were selected as the

The second modification is shown in **Line 203**.

We changed the title “CAPE showed good biocompatibility” to “The biocompatibility of CAPE”.

203 ***The biocompatibility of CAPE***[↵]

The third modification is shown in **Lines 208-211**.

We changed

“The viability of HOK cells was above 80% with 40 $\mu\text{g/mL}$ and 80 $\mu\text{g/mL}$ CAPE and above 90% on average with a CAPE concentration of 20 $\mu\text{g/mL}$, which indicated the good biocompatibility of CAPE.”

to

“The viability of HOK cells was above 80% with 40 $\mu\text{g/mL}$ and 80 $\mu\text{g/mL}$ CAPE and above 90% on average with a CAPE concentration of 20 $\mu\text{g/mL}$. Compared with CHX, which exhibited obvious cytotoxicity (see the Discussion section), the biocompatibility of CAPE is acceptable.”

208 slightly inhibited with the tested concentrations of CAPE. The viability of HOK cells was above
209 80% with 40 $\mu\text{g/mL}$ and 80 $\mu\text{g/mL}$ CAPE and above 90% on average with a CAPE concentration
210 of 20 $\mu\text{g/mL}$. Compared with CHX, which exhibited obvious cytotoxicity (see the Discussion
211 section), the biocompatibility of CAPE is acceptable.[↵]

In the Discussion,

We added “The biocompatibility of CAPE has been reported by previous studies (26).

In our research, CAPE showed slight cytotoxicity to human oral keratinocytes (HOKs), according to the classification (25), which rates extracts as having severe, moderate or slight cytotoxicity when the viability relative to the control groups is less than 30%, between 30% and 60%, or greater than 60%, respectively. It is worth noting that CHX is currently considered one of the most common antiseptics in dentistry and the gold standard agent for the application of oral rinses; in particular, the clinical

concentration of 0.12% CHX is used to remove plaque biofilm (21, 22, 27). However, its cytotoxicity has been widely reported, especially the obvious reduction of CHX against HOKs and fibroblast cell numbers in a short time (28). Barbara Azzimonti et al. (24) also emphasized that a low concentration of CHX (0.05%) resulted in less than 40% mucosal keratinocytes viability. Thus, our research suggested that the biocompatibility of CAPE is acceptable and has application potential.” in **Lines 291-302**.

The biocompatibility of CAPE has been reported by previous studies (21). In our
research, CAPE showed slight cytotoxicity to human oral keratinocytes (HOKs), according to the
classification (37), which rates extracts as having severe, moderate or slight cytotoxicity when
the viability relative to the control groups is less than 30%, between 30% and 60%, or greater
than 60%, respectively. It is worth noting that CHX is currently considered one of the most
common antiseptics in dentistry and the gold standard agent for the application of oral rinses, in
particular, the clinical concentration of 0.12% CHX is used to remove plaque biofilm (5, 38, 39).
However, its cytotoxicity has been widely reported, especially the obvious reduction of CHX
against HOKs and fibroblast cell numbers in a short time (40). Barbara Azzimonti et al. (41) also
emphasized that a low concentration of CHX (0.05%) resulted in less than 40% mucosal
keratinocytes viability. Thus, our research suggested that the biocompatibility of CAPE is
acceptable and has application potential.↵

Additionally, we changed “good” to “acceptable” in **Line 317**.

317 coinfections. CAPE also exhibited **acceptable** biocompatibility, but more extensive research for

Comment 8:

What is the MIC of CAPE for other strains of *S. mutans* and *C. albicans*. Are they similar to UA159 and SC5314? How did you pick these strains for the study?

Response 8:

Thank you very much for this question. The selection of strains was taken into account in the experimental design stage. We explain the process in detail as follows. First, both the *S. mutans* strain UA159 (ATCC 700610) and the *C. albicans* strain SC5314 (ATCC 10691) have been extensively applied as universal reference strains from American Type Culture Collection (ATCC). The reference strains have

comprehensive and typical microbiological characteristics, which can be used as the standard of quality control in microbiological examination and scientific research.

The genome of *S. mutans* strain UA159 has been sequenced (29). Based on the UA159 genome, studies with comparative genomic hybridization (CGH) have shown a high degree of content variation among strains, with some isolates lacking up to 20% of the genes present in the reference strain UA159 (30, 31). Thus, to make our research more comprehensive, standardized and valuable in application, the *S. mutans* strain UA159 was selected, which is consistent with other studies (17, 32, 33).

Additionally, the *C. albicans* strain SC5314 was used in this study as a reference strain and is known to be severely invasive with strong biofilm formation (34-36). This use is also in accordance with other reports (35-38), especially studies of the effects against biofilms.

Therefore, in our opinion, this study on the effects of CAPE on the *S. mutans* strain UA159 and the *C. albicans* strain SC5314 would be meaningful for clinical applications. Further effects of CAPE for other strains of *S. mutans* and *C. albicans* will be investigated in subsequent studies.

Comment 9:

How can a CFU be determined from a biofilm in Figure 2B? Don't they clump-up and form aggregates, giving false CFU?

Response 9:

Thank you very much for noting that we missed critical experimental steps in the Materials and Methods. We will explain this process clearly as follows.

In the CFU-counting assay, the biofilm aggregates were scraped off from the bottom of the microwells. According to reported protocols (39, 40), the samples were then mixed thoroughly by vortexing with sterile PBS so that the suspension could be used to dilute, incubate and count colonies.

Accordingly, we having provided this missing information and added references, as shown in **Lines 393-398**.

We changed

“The adherent biofilms were then removed from the floor of each microwell, resuspended in sterile PBS and diluted 10⁶-fold to count *S. mutans* colonies and 10⁴-fold to count *C. albicans* colonies.”

to

“The adherent biofilms were then removed from the floor of each microwell and thoroughly mixed by vortexing with 200 μ L of sterile PBS. After resuspension, each sample was diluted 10⁶-fold to count *S. mutans* colonies and 10⁴-fold to count *C. albicans* colonies.”

393 h, the biofilms were carefully washed with sterile PBS to eliminate planktonic cells. The
394 adherent biofilms were then removed from the floor of each microwell and thoroughly mixed by
395 vortexing with 200 μ L of sterile PBS. After resuspension, each sample was diluted 10⁶-fold to
396 count *S. mutans* colonies and 10⁴-fold to count *C. albicans* colonies. Colonies were counted
following incubation on BHI agar medium for *S. mutans* and on YPD agar medium for *C.*
*albicans* (17, 29, 51).⁴¹

Comment 10:

In Figure 5 and lines 182-188, authors need to explain what *gtf* genes do? And why they decide to test for the expression level of these genes? Did the authors do comprehensive screen for other cariogenic genes?

Response 10:

Thank you for this valuable suggestion. We will explain this aspect clearly.

Glucosyltransferases (Gtfs) play crucial roles in the occurrence and development of *S. mutans*-mediated caries, including early childhood caries (41). Generally, Gtfs can produce extracellular polysaccharides (EPS) to promote biofilm formation and mediate the adhesion of cariogenic bacteria (4, 42). The *gtf* genes (*gtfB*, *gtfC*, *gtfD*) encode three Gtf enzymes (GtfB, GtfC, GtfD) produced by *S. mutans*. The expression of the three *gtf* genes is distinct but related according to a reported study (41). Briefly, the expression level of *gtf* genes is closely associated with EPS synthesis.

Furthermore, at the beginning of the study design, we extensively screened cariogenic genes to determine the expression level of these three *gtf* genes, which are closely related to the cariogenicity of *S. mutans*. This finding is consistent with other studies on cariogenic genes (18, 43).

According to your suggestion, we have added the following text, as shown in **Lines 193-195**.

“Gtfs, as enzymes that ferment sugar, catalyzing the transformation of glucosyl groups, contribute to the synthesis of EPS by *S. mutans*. The expression level of *gtf* genes is closely related to EPS synthesis.”

Gtfs, as enzymes that ferment sugar, catalyzing the transformation of glucosyl groups, contribute
to the synthesis of EPS by *S. mutans*. The expression level of *gtf* genes is closely related to EPS
synthesis. The results of the qRT-PCR assay showed the relative expression levels of the *gtf*

Additionally, we revised the text in **Lines 199-201** as follows.

We changed “cariogenic genes” to “*gtf* genes (*gtfB*, *gtfC* and *gtfD*)”.

influence of *C. albicans* in the cross-kingdom biofilm (Fig. 6B). Generally, the statistical results
revealed the inhibitory effect of CAPE on the expression of *gtf* genes (*gtfB*, *gtfC* and *gtfD*) in *S.*
*mutans*.↵

Comment 11:

It will go a long way to show additional oral epithelial cell tests. HOK cells are one of many oral keratinocytes and they may show different effect compare to other cell lines.

Response 11:

We appreciate your valuable comment; we will explain our consideration as follows.

Oral epithelial cells can be divided into keratinocytes and nonkeratinocytes according to whether they participate in keratinization. Oral keratinocytes, which were used in our study, act as the major barrier to oral diseases, protecting local cells from physical, microbial, and chemical damage. Furthermore, oral keratinocytes can potentially participate in controlling oral infections through the inflammatory process according to the report (44). Consequently, in our opinion, the application of oral keratinocytes for cell viability research can be representative, which is consistent with other studies on the effects of natural products or extracts against oral microbes (22, 45, 46).

Subsequently, based on the measurement of cell viability as the method of cytotoxicity evaluation, we will conduct a more comprehensive assessment of CAPE’s effects on inflammatory reactions in animal or clinical models. This aspect deserves to be investigated for more comprehensive understanding of biocompatibility in our subsequent studies, which is consistent with previous reports (47, 48).

We would like to take this opportunity to thank you for all your time involved and this opportunity for us to improve the manuscript. We hope you will find this revised

version satisfactory.

Sincerely,

Derong Yin

References

1. Hajishengallis E, Parsaei Y, Klein MI, Koo H. 2017. Advances in the microbial etiology and pathogenesis of early childhood caries. *Mol Oral Microbiol* 32:24-34.
2. Tanner ACR, Kressirer CA, Rothmiller S, Johansson I, Chalmers NI. 2018. The Caries Microbiome: Implications for Reversing Dysbiosis. *Adv Dent Res* 29:78-85.
3. Hwang G, Liu Y, Kim D, Li Y, Krysan DJ, Koo H. 2017. *Candida albicans* mannans mediate *Streptococcus mutans* exoenzyme GtfB binding to modulate cross-kingdom biofilm development in vivo. *PLoS Pathog* 13:e1006407.
4. Niu Y, Wang K, Zheng S, Wang Y, Ren Q, Li H, Ding L, Li W, Zhang L. 2020. Antibacterial Effect of Caffeic Acid Phenethyl Ester on Cariogenic Bacteria and *Streptococcus mutans* Biofilms. *Antimicrob Agents Chemother* 64.
5. de Barros PP, Rossoni RD, Garcia MT, Kaminski VL, Loures FV, Fuchs BB, Mylonakis E, Junqueira JC. 2021. The Anti-Biofilm Efficacy of Caffeic Acid Phenethyl Ester (CAPE) In Vitro and a Murine Model of Oral Candidiasis. *Front Cell Infect Microbiol* 11:700305.
6. Olgierd B, Kamila Z, Anna B, Emilia M. 2021. The Pluripotent Activities of Caffeic Acid Phenethyl Ester. *Molecules* 26.
7. Armutcu F, Akyol S, Ustunsoy S, Turan FF. 2015. Therapeutic potential of caffeic acid phenethyl ester and its anti-inflammatory and immunomodulatory effects (Review).

Exp Ther Med 9:1582-1588.

8. Alfarrayeh I, Pollak E, Czeh A, Vida A, Das S, Papp G. 2021. Antifungal and Anti-Biofilm Effects of Caffeic Acid Phenethyl Ester on Different Candida Species. Antibiotics (Basel) 10.
9. Veloz JJ, Alvear M, Salazar LA. 2019. Antimicrobial and Antibiofilm Activity against Streptococcus mutans of Individual and Mixtures of the Main Polyphenolic Compounds Found in Chilean Propolis. Biomed Res Int 2019:7602343.
10. Sun L, Liao K, Hang C. 2018. Caffeic acid phenethyl ester synergistically enhances the antifungal activity of fluconazole against resistant Candida albicans. Phytomedicine 40:55-58.
11. Masadeh MM, Gharaibeh SF, Alzoubi KH, Al-Azzam SI, Obeidat WM. 2013. Antimicrobial activity of common mouthwash solutions on multidrug-resistance bacterial biofilms. J Clin Med Res 5:389-94.
12. Rajendiran M, Trivedi HM, Chen D, Gajendrareddy P, Chen L. 2021. Recent Development of Active Ingredients in Mouthwashes and Toothpastes for Periodontal Diseases. Molecules 26.
13. Jhingta P, Bhardwaj A, Sharma D, Kumar N, Bhardwaj VK, Vaid S. 2013. Effect of hydrogen peroxide mouthwash as an adjunct to chlorhexidine on stains and plaque. J Indian Soc Periodontol 17:449-53.
14. Aljuffali IA, Fang CL, Chen CH, Fang JY. 2016. Nanomedicine as a Strategy for Natural Compound Delivery to Prevent and Treat Cancers. Curr Pharm Des 22:4219-31.

15. Shih YH, Hsia SM, Chiu KC, Wang TH, Chien CY, Li PJ, Kuo YH, Shieh TM. 2022. In Vitro Antimicrobial Potential of CAPE and Caffeamide Derivatives against Oral Microbes. *Int J Mol Sci* 23.
16. Tambuwala MM, Khan MN, Thompson P, McCarron PA. 2019. Albumin nano-encapsulation of caffeic acid phenethyl ester and piceatannol potentiated its ability to modulate HIF and NF- κ B pathways and improves therapeutic outcome in experimental colitis. *Drug Deliv Transl Res* 9:14-24.
17. Zhang Z, Liu Y, Lu M, Lyu X, Gong T, Tang B, Wang L, Zeng J, Li Y. 2020. *Rhodiola rosea* extract inhibits the biofilm formation and the expression of virulence genes of cariogenic oral pathogen *Streptococcus mutans*. *Arch Oral Biol* 116:104762.
18. Zhang Z, Lyu X, Xu Q, Li C, Lu M, Gong T, Tang B, Wang L, Zeng W, Li Y. 2020. Utilization of the extract of *Cedrus deodara* (Roxb. ex D.Don) G. Don against the biofilm formation and the expression of virulence genes of cariogenic bacterium *Streptococcus mutans*. *J Ethnopharmacol* 257:112856.
19. Lee WH, Pressman JG, Wahman DG. 2018. Three-Dimensional Free Chlorine and Monochloramine Biofilm Penetration: Correlating Penetration with Biofilm Activity and Viability. *Environ Sci Technol* 52:1889-1898.
20. Morris AJ, Li A, Jackson L, Yau YCW, Waters V. 2020. Quantifying the Effects of Antimicrobials on In vitro Biofilm Architecture using COMSTAT Software. *J Vis Exp.* <https://doi.org/10.3791/61759>.
21. Karpinski TM, Szkaradkiewicz AK. 2015. Chlorhexidine--pharmaco-biological activity and application. *Eur Rev Med Pharmacol Sci* 19:1321-6.

22. Oliveira MAC, Borges AC, Brighenti FL, Salvador MJ, Gontijo AVL, Koga-Ito CY. 2017. Cymbopogon citratus essential oil: effect on polymicrobial caries-related biofilm with low cytotoxicity. Braz Oral Res 31:e89.
23. Shino B, Peedikayil FC, Jaiprakash SR, Ahmed Bijapur G, Kottayi S, Jose D. 2016. Comparison of Antimicrobial Activity of Chlorhexidine, Coconut Oil, Probiotics, and Ketoconazole on Candida albicans Isolated in Children with Early Childhood Caries: An In Vitro Study. Scientifica (Cairo) 2016:7061587.
24. Azzimonti B, Cochis A, Beyrouthy ME, Iriti M, Uberti F, Sorrentino R, Landini MM, Rimondini L, Varoni EM. 2015. Essential Oil from Berries of Lebanese Juniperus excelsa M. Bieb Displays Similar Antibacterial Activity to Chlorhexidine but Higher Cytocompatibility with Human Oral Primary Cells. Molecules 20:9344-57.
25. Sletten GB, Dahl JE. 1999. Cytotoxic effects of extracts of compomers. Acta Odontol Scand 57:316-22.
26. Olgierd B, Kamila Z, Anna B, Emilia M. 2021. The pluripotent activities of caffeic acid phenethyl ester. Molecules 26:1335.
27. Niu Y, Wang K, Zheng S, Wang Y, Ren Q, Li H, Ding L, Li W, Zhang L. 2020. Antibacterial effect of caffeic acid phenethyl ester on cariogenic bacteria and Streptococcus mutans biofilms. Antimicrob Agents Chemother 64:e00251-20.
28. Balloni S, Locci P, Lumare A, Marinucci L. 2016. Cytotoxicity of three commercial mouthrinses on extracellular matrix metabolism and human gingival cell behaviour. Toxicol In Vitro 34:88-96.
29. Ajdic D, McShan WM, McLaughlin RE, Savic G, Chang J, Carson MB, Primeaux C,

- Tian R, Kenton S, Jia H, Lin S, Qian Y, Li S, Zhu H, Najjar F, Lai H, White J, Roe BA, Ferretti JJ. 2002. Genome sequence of *Streptococcus mutans* UA159, a cariogenic dental pathogen. *Proc Natl Acad Sci U S A* 99:14434-9.
30. Waterhouse JC, Swan DC, Russell RR. 2007. Comparative genome hybridization of *Streptococcus mutans* strains. *Oral Microbiol Immunol* 22:103-10.
 31. Zhang L, Foxman B, Drake DR, Srinivasan U, Henderson J, Olson B, Marrs CF, Warren JJ, Marazita ML. 2009. Comparative whole-genome analysis of *Streptococcus mutans* isolates within and among individuals of different caries status. *Oral Microbiol Immunol* 24:197-203.
 32. Morrison DA, Khan R, Junges R, Amdal HA, Petersen FC. 2015. Genome editing by natural genetic transformation in *Streptococcus mutans*. *J Microbiol Methods* 119:134-41.
 33. Hale JD, Heng NC, Jack RW, Tagg JR. 2005. Identification of nImTE, the locus encoding the ABC transport system required for export of nonantibiotic mutacins in *Streptococcus mutans*. *J Bacteriol* 187:5036-9.
 34. Parmanen P, Meurman JH, Samaranayake L, Virtanen I. 2010. Human oral keratinocyte E-cadherin degradation by *Candida albicans* and *Candida glabrata*. *J Oral Pathol Med* 39:275-8.
 35. Baron G, Altomare A, Regazzoni L, Fumagalli L, Artasensi A, Borghi E, Ottaviano E, Del Bo C, Riso P, Allegrini P, Petrangolini G, Morazzoni P, Riva A, Arnoldi L, Carini M, Aldini G. 2020. Profiling *Vaccinium macrocarpon* components and metabolites in human urine and the urine ex-vivo effect on *Candida albicans* adhesion and

biofilm-formation. *Biochem Pharmacol* 173:113726.

36. Khan MS, Ahmad I. 2012. Antibiofilm activity of certain phytochemicals and their synergy with fluconazole against *Candida albicans* biofilms. *J Antimicrob Chemother* 67:618-21.
37. Bakri MM, Rich AM, Cannon RD, Holmes AR. 2015. In vitro expression of *Candida albicans* alcohol dehydrogenase genes involved in acetaldehyde metabolism. *Mol Oral Microbiol* 30:27-38.
38. Chang FM, Ou TY, Cheng WN, Chou ML, Lee KC, Chin YP, Lin CP, Chang KD, Lin CT, Su CH. 2014. Short-term exposure to fluconazole induces chromosome loss in *Candida albicans*: an approach to produce haploid cells. *Fungal Genet Biol* 70:68-76.
39. Zhang N, Chen C, Melo MA, Bai YX, Cheng L, Xu HH. 2015. A novel protein-repellent dental composite containing 2-methacryloyloxyethyl phosphorylcholine. *Int J Oral Sci* 7:103-9.
40. Liu S, Qiu W, Zhang K, Zhou X, Ren B, He J, Xu X, Cheng L, Li M. 2017. Nicotine Enhances Interspecies Relationship between *Streptococcus mutans* and *Candida albicans*. *Biomed Res Int* 2017:7953920.
41. Zhang Q, Ma Q, Wang Y, Wu H, Zou J. 2021. Molecular mechanisms of inhibiting glucosyltransferases for biofilm formation in *Streptococcus mutans*. *Int J Oral Sci* 13:30.
42. Bowen WH, Burne RA, Wu H, Koo H. 2018. Oral Biofilms: Pathogens, Matrix, and Polymicrobial Interactions in Microenvironments. *Trends Microbiol* 26:229-242.
43. Xu X, Zhou XD, Wu CD. 2012. Tea catechin epigallocatechin gallate inhibits

Streptococcus mutans biofilm formation by suppressing gtf genes. Arch Oral Biol 57:678-83.

44. Groeger S, Meyle J. 2019. Oral Mucosal Epithelial Cells. Front Immunol 10:208.
45. Zocolotti JO, Cavalheiro AJ, Tasso CO, Ribas BR, Ferrisse TM, Jorge JH. 2021. Antimicrobial efficacy and biocompatibility of extracts from Cryptocarya species. PLoS One 16:e0261884.
46. Vaillancourt K, LeBel G, Pellerin G, Ben Lagha A, Grenier D. 2021. Effects of the Licorice Isoflavans Licoricidin and Glabridin on the Growth, Adherence Properties, and Acid Production of Streptococcus mutans, and Assessment of Their Biocompatibility. Antibiotics (Basel) 10.
47. Giusto G, Beretta G, Vercelli C, Valle E, Iussich S, Borghi R, Odetti P, Monacelli F, Tramuta C, Grego E, Nebbia P, Robino P, Odore R, Gandini M. 2018. Pectin-honey hydrogel: Characterization, antimicrobial activity and biocompatibility. Biomed Mater Eng 29:347-356.
48. Kwon JS, Piao YZ, Cho SA, Yang SY, Kim JH, An S, Kim KM. 2015. Biocompatibility Evaluation of Dental Luting Cements Using Cytokine Released from Human Oral Fibroblasts and Keratinocytes. Materials (Basel) 8:7269-7277.

Response to Reviewer #3

Dear Reviewer,

We sincerely thank you for your professional review work and your valuable feedback, which we have used to improve the quality of our manuscript. According to your kind suggestions, we have made extensive modifications to our manuscript, and the detailed corrections are listed. Please see below, in blue text, our point-by-point responses to your comments and concerns. All page numbers refer to the revised manuscript file with tracked changes.

Comment 1:

Abstract: ASM Journals not use sectioned abstracts, please combine and reword accordingly.

Response 1:

Thank you for the suggestion. We have revised the Abstract as follows (Lines 19-37).

**Abstract**[↵]
*Streptococcus mutans* and *Candida albicans* exhibit strong cariogenicity through cross-kingdom
biofilm formation during the pathogenesis of dental caries. Caffeic acid phenethyl ester (CAPE),
a natural compound, has potential antimicrobial effects on each species individually, but there are
no reports of its effect on this dual-species biofilm. This study aimed to explore the effects of
CAPE on cariogenic biofilm formation by *S. mutans* and *C. albicans* and the related
mechanisms. The effect of CAPE on planktonic cell growth was investigated, and crystal violet
staining, the anthrone-sulfuric acid assay and confocal laser scanning microscopy (CLSM) were
used to evaluate biofilm formation. The structures of the formed biofilms were observed using
scanning electron microscopy. To explain the antimicrobial effect of CAPE, quantitative real-
time PCR (qRT-PCR) was applied to monitor the relative expression levels of cariogenic genes.
Finally, the biocompatibility of CAPE in human oral keratinocytes (HOKs) was evaluated using
the CCK-8 assay. The results showed that CAPE suppressed the growth, biofilm formation and
extracellular polysaccharides (EPS) synthesis of *C. albicans* and *S. mutans* in the co-culture
system of the two species. The expression of the *gtf* gene was also suppressed by CAPE. The
efficacy of CAPE was concentration dependent, and the compound exhibited acceptable
biocompatibility. Our research lays the foundation for further study of the application of the
natural compound CAPE as a potential antimicrobial agent to control dental caries-associated
cross-kingdom biofilms.[↵]

Comment 2:

Importance: "closely-related to the cross-kingdom", this sentence has awkward phrasing as currently written, please revise.

Response 2:

Thank you for your comment. We have modified this sentence to make the meaning more accurate and clearer. The revised text reads as follows (**Lines 39-40**).

We changed

“Severe dental caries is closely related to the cross-kingdom biofilm formed by *S. mutans* and *C. albicans*.”

to

“Severe dental caries is a multimicrobial infectious disease that is strongly induced by the cross-kingdom biofilm formed by *S. mutans* and *C. albicans*.”

Severe dental caries is a multimicrobial infectious disease that is strongly induced by the cross-
kingdom biofilm formed by *S. mutans* and *C. albicans*. This study aimed to investigate the

Comment 3:

Line 57: please change "the key pathogen" to "a key pathogen", as many researchers in the field would consider this an exaggeration of the role of *S. mutans*, as cases of caries without detectable levels of *S. mutans* are routine.

Response 3:

Thank you for your careful reading and kind reminder. We completely agree that it is inaccurate to regard *S. mutans* as the key pathogen of dental caries. Thus, we have made the modification according to your opinion, as shown in **Line 55**.

Considered a key pathogen, *Streptococcus mutans* is involved in the occurrence and

Comment 4:

Line 110: "corresponding colony variations" please clarify or change? Do you mean number of CFUs or differences in the actual colony morphology?

Response 4:

Thank you for noting this issue. We were attempting to emphasize the changes in the number of colonies, as shown by the CFU assay.

Therefore, we have corrected this inaccurate text as follows (**Line 110**).

We changed “corresponding colony variations” to “corresponding variations in colony numbers”.

examine the corresponding variations of colony numbers in *S. mutans* and *C. albicans* under

Comment 5:

Line 156: "was affected by CAPE" awkward phrasing again here...I think you mean something more like "a reduction in biofilm accumulation correlated with increasing concentration of CAPE"

Response 5:

We gratefully appreciate your careful review and noting our inaccurate statement. We apologize for the misunderstanding caused by awkward phrasing.

According to your suggestion, we made the modification shown in **Lines 156-157**.

We changed "the observable reduction in biofilm accumulation was affected by CAPE." to "the observable reduction in biofilm accumulation correlated with increasing CAPE concentrations."

Similarly, for the cross-kingdom biofilms shown in Fig. 3C, it is clear that the observable
reduction in biofilm accumulation correlated with increasing CAPE concentrations. The

Comment 6:

qRT-PCR: This is important! If I understand the methods correctly, the relative expression of *gtfB*, *C*, and *D* in each sample is "relative" to the corresponding 0 μ g CAPE sample. Since there are clearly fewer cells, and more dead cells in the higher CAPE concentration samples, you would expect less *gtf* RNA, and indeed less RNA overall in those samples. Crucially, using this method, you therefore cannot hypothesize that CAPE specifically reduced the transcription of *gtf* genes. You would need to control for the differences in cell viability and/or also compare to the RNA levels of other genes. Please either add experiments with the proper controls or remove this experiment from the study.

Response 6:

Thank you for noting this issue; we will explain this method.

In our research, *S. mutans* UA159 16S rRNA was used as an internal control, according to previous studies (1-3). It has been reported that the bacterial 16S rRNA gene, as the reference gene, can be applied to calculate the relative abundance of the specific 16S rRNA genes of the target bacteria and the relative expression level of the functional genes (4). The relative expression level of functional genes was simply

defined as the ratio of DNA fragments detected to the total bacteria detected via qPCR.

Accordingly, our research utilized *S. mutans* UA159 16S rRNA as the reference to calculate the relative expression level of the *gtf* genes (*gtfB*, *gtfC*, *gtfD*), both in the control group without CAPE treatment and in CAPE-treated groups with different concentrations. Through the measurement of real-time qPCR, the results could be interpreted as the average amount of target gene expression per bacterium, which was defined as the relative mRNA expression levels of *gtf* genes.

We apologize for the misunderstanding caused by the inaccurate statement. Therefore, we made the modification shown in **Lines 463-464**.

We changed

“The expression levels of *gtfB*, *gtfC* and *gtfD* mRNA were determined by the qRT-PCR assay, with *S. mutans* UA159 used as the reference.”

to

“The relative expression levels of *gtfB*, *gtfC* and *gtfD* mRNA were determined using the qRT-PCR assay, with *S. mutans* UA159 16S rRNA used as the reference.”

(TaKaRa, Japan). The relative expression levels of *gtfB*, *gtfC* and *gtfD* mRNA were determined
using the qRT-PCR assay, with *S. mutans* UA159 16S rRNA used as the reference. The primers

Comment 7:

Biocompatibility: It is not clear from the current manuscript what current "acceptable" levels of CCK-8 viability are, and how much toxicity in the assay might translate to toxicity in vivo. Please either remove claims of "good biocompatibility" or add references explaining what "good biocompatibility" is and how the results here compare to it.

Response 7:

Thank you very much for noting this issue.

According to your valuable suggestions, we chose chlorhexidine (CHX) as the positive control in our research. CHX is considered the gold standard substance for application in oral rinses and has been extensively used to remove plaque biofilms in clinical dentistry; in particular, a 0.12% concentration of CHX is used to prevent the occurrence and progression of caries (5-7). However, its cytotoxicity remains a disadvantage that has not been well resolved (8). According to reported research (9), a low concentration of CHX (0.05%) resulted in oral keratinocyte viability of less than

40%.

In our research, at a 20 µg/mL concentration of CAPE, human oral keratinocytes (HOKs) exhibited approximately 90% viability. Additionally, the HOK cells had more than 80% viability at 40 µg/mL and 80 µg/mL (MIC). According to the classification of cytotoxicity (10), extracts are considered severely, moderately or slightly cytotoxic when the viability relative to the controls is less than 30%, between 30% and 60%, or greater than 60%, respectively. Therefore, CAPE illustrated “slight cytotoxicity” in our research instead of good biocompatibility, which has been corrected as shown in **Line 292**.

research, CAPE showed slight cytotoxicity to human oral keratinocytes (HOKs), according to the

Although all CAPE-treated groups showed a significant difference from the control group, the 80% oral keratinocyte viability with CAPE treatment was significantly higher than that of cells treated with a low concentration of CHX (below 40% viability). Additionally, our supplementary susceptibility assay data show that the MIC of CHX ranged from 4×10^3 to 8×10^3 mg/mL for *S. mutans* and *C. albicans*, which was significantly higher than the MIC concentration of CAPE. Therefore, our research suggested that the biocompatibility of CAPE is acceptable and has application potential.

Accordingly, we have made the supplements and modifications in the Abstract, Materials and Methods, Results, and Discussion.

In the Abstract, we changed “good” to “acceptable” in **Lines 34-35**.

34 efficacy of CAPE was concentration dependent, and the compound exhibited acceptable

35 biocompatibility. Our research lays the foundation for further study of the application of the

In Materials and Methods, as shown in **Lines 342-344**.

We changed

“CHX (Sigma–Aldrich, Steinheim, Germany) replaced CAPE in microcells as the positive control according to reported protocols”

to

“As the positive control, CHX (Sigma–Aldrich, Steinheim, Germany) replaced CAPE in microcells at concentrations ranging from 1.0×10^3 to 128.0×10^3 mg/mL according

to reported protocols.”

used as the solvent and blank controls, respectively. As the positive control, CHX (Sigma–
Aldrich, Steinheim, Germany) replaced CAPE in microcells at concentrations ranging from
1.0×10^3 to 128.0×10^3 mg/mL according to reported protocols (5). The microplates were

In the Results section, there are three changes.

The first addition is in **Lines 103-105**.

We added “The MIC of CHX ranged from 4×10^3 to 8×10^3 mg/mL for both strains as the positive control, which was significantly higher than the MIC concentration of CAPE.”

control had no effect on bacterial and fungal growth. The MIC of CHX ranged from 4×10^3 to
8×10^3 mg/mL for both strains as the positive control, which was significantly higher than the
MIC concentration of CAPE. Thus, 20, 40, 80 and 160 $\mu\text{g/mL}$ CAPE were selected as the

The second modification is shown in **Line 203**.

We changed the title “CAPE showed good biocompatibility” to “The biocompatibility of CAPE”.

***The biocompatibility of CAPE***[⚡]

The third modification is in **Lines 208-211**.

We changed

“The viability of HOK cells was above 80% with 40 $\mu\text{g/mL}$ and 80 $\mu\text{g/mL}$ CAPE and above 90% on average with a CAPE concentration of 20 $\mu\text{g/mL}$, which indicated the good biocompatibility of CAPE.”

to

“The viability of HOK cells was above 80% with 40 $\mu\text{g/mL}$ and 80 $\mu\text{g/mL}$ CAPE and above 90% on average with a CAPE concentration of 20 $\mu\text{g/mL}$. Compared with CHX, which exhibited obvious cytotoxicity (see the Discussion section), the biocompatibility of CAPE is acceptable.”

slightly inhibited with the tested concentrations of CAPE. The viability of HOK cells was above
80% with 40 $\mu\text{g/mL}$ and 80 $\mu\text{g/mL}$ CAPE and above 90% on average with a CAPE concentration
of 20 $\mu\text{g/mL}$. Compared with CHX, which exhibited obvious cytotoxicity (see the Discussion
section), the biocompatibility of CAPE is acceptable.[⚡]

In the Discussion,

We added “The biocompatibility of CAPE has been reported by previous studies (11). In our research, CAPE showed slight cytotoxicity to human oral keratinocytes (HOKs), according to the classification (10), which rates extracts as having severe, moderate or slight cytotoxicity when the viability relative to the control groups is less than 30%, between 30% and 60%, or greater than 60%, respectively. It is worth noting that CHX is currently considered one of the most common antiseptics in dentistry and the gold standard agent for the application of oral rinses, in particular, the clinical concentration of 0.12% CHX is used to remove plaque biofilm (6, 7, 12). However, its cytotoxicity has been widely reported, especially the obvious reduction of CHX against HOKs and fibroblast cell numbers in a short time (13). Barbara Azzimonti et al. (9) also emphasized that a low concentration of CHX (0.05%) resulted in less than 40% mucosal keratinocytes viability. Thus, our research suggested that the biocompatibility of CAPE is acceptable and has application potential.” in **Lines 291-302**.

The biocompatibility of CAPE has been reported by previous studies (21). In our
research, CAPE showed slight cytotoxicity to human oral keratinocytes (HOKs), according to the
classification (37), which rates extracts as having severe, moderate or slight cytotoxicity when
the viability relative to the control groups is less than 30%, between 30% and 60%, or greater
than 60%, respectively. It is worth noting that CHX is currently considered one of the most
common antiseptics in dentistry and the gold standard agent for the application of oral rinses, in
particular, the clinical concentration of 0.12% CHX is used to remove plaque biofilm (5, 38, 39).
However, its cytotoxicity has been widely reported, especially the obvious reduction of CHX
against HOKs and fibroblast cell numbers in a short time (40). Barbara Azzimonti et al. (41) also
emphasized that a low concentration of CHX (0.05%) resulted in less than 40% mucosal
keratinocytes viability. Thus, our research suggested that the biocompatibility of CAPE is
acceptable and has application potential.↵

Additionally, we changed “good” to “acceptable” in **Line 317**.

317 coinfections. CAPE also exhibited acceptable biocompatibility, but more extensive research for

Comment 8:

Line 218: please remove "promising", I think that's a bit overboard at this stage.

Response 8:

Thank you for the detailed review, and we agree with your suggestion.

Therefore, we changed “this promising natural product” to “this natural product”, as shown in **Lines 231-232**.

interfere with caries infection caused by single pathogenic bacteria (5), the use of **this natural**
**product** remains to be explored for preventing cross-kingdom biofilms with synergistic

Comment 9:

Lines 251-257: see comment #6 above

Response 9:

Thank you again for noting the discussion about the expression of cariogenic gene by qRT-PCR. We hope that response #6 above can resolve these important and valuable questions.

We have also modified the inaccurate text, as shown in **Lines 265-271**.

abundantly in cariogenic bacteria. Based on the outcomes of the ~~qRT-PCR~~ assay, we speculate
that the inhibitory effect of CAPE on the biofilm and EPS is **probably** achieved by reducing the
expression level of *gtf* genes, including *gtfB*, *gtfC*, and *gtfD*. Notably, the expression of *gtfB*
contributes to the adhesion of *S. mutans* and *C. albicans* (11, 26, 31). As shown in other studies
in vitro, *C. albicans* increased the formation of the extracellular matrix of *S. mutans* by
upregulating the expression levels of *gtfB* and *gtfC* in the cross-kingdom biofilm with symbiosis
of these two species (26, 32). Therefore, CAPE downregulated the relative expression of the *gtf*

Comment 10:

Line 265: do you mean "plants" instead of "implants"?

Response 10:

We apologize for our careless error; thank you so much for your comment.

We have corrected “implants” to “plants”, as shown in **Line 279**.

flavonoids, cinnamaldehyde, ~~cinnamol~~ and other natural compounds extracted from various
**plants** have been **indicated** to be effective antibacterial agents against *S. mutans* (17, 34, 35).

Comment 11:

Line 277: see comment #7 above.

Response 11:

Thank you again for noting our inaccurate description of the biocompatibility of CAPE. According to your suggestions, as shown in comment #6 above, we have added “The biocompatibility of CAPE has been reported by previous studies (11). In our research, CAPE showed slight cytotoxicity to human oral keratinocytes (HOKs), according to the classification (10), which rates extracts as having severe, moderate or slight cytotoxicity when the viability relative to the control groups is less than 30%, between 30% and 60%, or greater than 60%, respectively. It is worth noting that CHX is currently considered one of the most common antiseptics in dentistry and the gold standard agent for the application of oral rinses; in particular, the clinical concentration of 0.12% CHX is used to remove plaque biofilm (6, 7, 12). However, its cytotoxicity has been widely reported, especially the obvious reduction of CHX against HOKs and fibroblast cell numbers in a short time (13). Barbara Azzimonti et al. (9) also emphasized that a low concentration of CHX (0.05%) resulted in less than 40% mucosal keratinocytes viability. Thus, our research suggested that the biocompatibility of CAPE is acceptable and has application potential.” in **Lines 291-302**.

The biocompatibility of CAPE has been reported by previous studies (21). In our
research, CAPE showed slight cytotoxicity to human oral keratinocytes (HOKs), according to the
classification (37), which rates extracts as having severe, moderate or slight cytotoxicity when
the viability relative to the control groups is less than 30%, between 30% and 60%, or greater
than 60%, respectively. It is worth noting that CHX is currently considered one of the most
common antiseptics in dentistry and the gold standard agent for the application of oral rinses, in
particular, the clinical concentration of 0.12% CHX is used to remove plaque biofilm (5, 38, 39).
However, its cytotoxicity has been widely reported, especially the obvious reduction of CHX
against HOKs and fibroblast cell numbers in a short time (40). Barbara Azzimonti et al. (41) also
emphasized that a low concentration of CHX (0.05%) resulted in less than 40% mucosal
keratinocytes viability. Thus, our research suggested that the biocompatibility of CAPE is
acceptable and has application potential.↵

Comment 12:

Lines 371-372: Since I don't believe that YPD and BHI are selective, how do you differentiate between colonies of *S. mutans* and *C. albicans* by plating on solid media from the co-culture? This is unclear, please explain.

Response 12:

We sincerely thank you for this important comment, which we also considered at the beginning of our research design.

In the CFU-counting assay for planktonic *S. mutans* and *C. albicans*, the cocultured suspensions were diluted 10⁴-fold to count *C. albicans* colonies, while *S. mutans* could not be recognized as having a colony morphology. Conversely, *S. mutans* colonies were counted with 10⁶-fold dilution of the cocultured suspensions, while the larger *C. albicans* cells rarely existed in the diluent. According to a reported study (3), this method for CFU counting was applied in our research, as shown in **Lines 357-362**. Therefore, we added the corresponding reference.

control group was treated with the solvent rather than CAPE. After incubation for 24 h, the
mixed suspensions were diluted 1:10⁶ (for counting *S. mutans*) and 1:10⁴ (for counting *C.*
*albicans*) with sterile phosphate-buffered saline (PBS) **in accordance with the reported study**
**(29)**. Then, *C. albicans* and *S. mutans* were incubated on YPD agar medium plates and BHI agar
medium plates, respectively, under anaerobic conditions with 5% CO₂ at 37°C for 48 h. The
colonies formed on the plates were then counted.[↵]

29. Liu S, Qiu W, Zhang K, Zhou X, Ren B, He J, Xu X, Cheng L, Li M. 2017. Nicotine Enhances Interspecies Relationship between *Streptococcus mutans* and *Candida albicans*. *Biomed Res Int* 2017:7953920.[↵]

Furthermore, in the CFU-counting assay for quantification of biofilm biomass, the biofilm aggregates were scraped off from the bottom of the microwells. In accordance with reported protocols (3, 14), the aggregates were then mixed thoroughly by vortexing with sterile PBS so that the suspension could be used to dilute, incubate and count colonies with the same method. As shown in **Lines 393-398**, we have provided more detailed text and added these references.

We changed

“The adherent biofilms were then removed from the floor of each microwell, resuspended in sterile PBS and diluted 10⁶-fold to count *S. mutans* colonies and 10⁴-fold to count *C. albicans* colonies.”

to

“The adherent biofilms were then removed from the floor of each microwell and thoroughly mixed by vortexing with 200 μL of sterile PBS. After resuspension, each sample was diluted 10⁶-fold to count *S. mutans* colonies and 10⁴-fold to count *C. albicans* colonies.”

393 h, the biofilms were carefully washed with sterile PBS to eliminate planktonic cells. The
394 adherent biofilms were then removed from the floor of each microwell and thoroughly mixed by
395 vortexing with 200 μ L of sterile PBS. After resuspension, each sample was diluted 10⁶-fold to
396 count *S. mutans* colonies and 10⁴-fold to count *C. albicans* colonies. Colonies were counted
following incubation on BHI agar medium for *S. mutans* and on YPD agar medium for *C.*
*albicans* (17, 29, 51).⁴

We would like to take this opportunity to thank you for your time and for this opportunity for us to improve the manuscript. We hope you will find this revised version satisfactory.

Sincerely,

Derong Yin

References

1. Zhang Z, Lyu X, Xu Q, Li C, Lu M, Gong T, Tang B, Wang L, Zeng W, Li Y. 2020. Utilization of the extract of *Cedrus deodara* (Roxb. ex D.Don) G. Don against the biofilm formation and the expression of virulence genes of cariogenic bacterium *Streptococcus mutans*. *J Ethnopharmacol* 257:112856.
2. Zhang Z, Liu Y, Lu M, Lyu X, Gong T, Tang B, Wang L, Zeng J, Li Y. 2020. *Rhodiola rosea* extract inhibits the biofilm formation and the expression of virulence genes of cariogenic oral pathogen *Streptococcus mutans*. *Arch Oral Biol* 116:104762.
3. Liu S, Qiu W, Zhang K, Zhou X, Ren B, He J, Xu X, Cheng L, Li M. 2017. Nicotine Enhances Interspecies Relationship between *Streptococcus mutans* and *Candida albicans*. *Biomed Res Int* 2017:7953920.
4. Zheng H, Xie T, Li S, Qiao X, Lu Y, Feng Y. 2021. Analysis of oral microbial dysbiosis

associated with early childhood caries. *BMC Oral Health* 21:181.

5. Niu Y, Wang K, Zheng S, Wang Y, Ren Q, Li H, Ding L, Li W, Zhang L. 2020. Antibacterial Effect of Caffeic Acid Phenethyl Ester on Cariogenic Bacteria and *Streptococcus mutans* Biofilms. *Antimicrob Agents Chemother* 64.
6. Karpinski TM, Szkaradkiewicz AK. 2015. Chlorhexidine--pharmaco-biological activity and application. *Eur Rev Med Pharmacol Sci* 19:1321-6.
7. Oliveira MAC, Borges AC, Brighenti FL, Salvador MJ, Gontijo AVL, Koga-Ito CY. 2017. *Cymbopogon citratus* essential oil: effect on polymicrobial caries-related biofilm with low cytotoxicity. *Braz Oral Res* 31:e89.
8. Shino B, Peedikayil FC, Jaiprakash SR, Ahmed Bijapur G, Kottayi S, Jose D. 2016. Comparison of Antimicrobial Activity of Chlorhexidine, Coconut Oil, Probiotics, and Ketoconazole on *Candida albicans* Isolated in Children with Early Childhood Caries: An In Vitro Study. *Scientifica (Cairo)* 2016:7061587.
9. Azzimonti B, Cochis A, Beyrouthy ME, Iriti M, Uberti F, Sorrentino R, Landini MM, Rimondini L, Varoni EM. 2015. Essential Oil from Berries of Lebanese *Juniperus excelsa* M. Bieb Displays Similar Antibacterial Activity to Chlorhexidine but Higher Cytocompatibility with Human Oral Primary Cells. *Molecules* 20:9344-57.
10. Sletten GB, Dahl JE. 1999. Cytotoxic effects of extracts of compomers. *Acta Odontol Scand* 57:316-22.
11. Olgierd B, Kamila Z, Anna B, Emilia M. 2021. The pluripotent activities of caffeic acid phenethyl ester. *Molecules* 26:1335.
12. Niu Y, Wang K, Zheng S, Wang Y, Ren Q, Li H, Ding L, Li W, Zhang L. 2020.

Antibacterial effect of caffeic acid phenethyl ester on cariogenic bacteria and *Streptococcus mutans* biofilms. *Antimicrob Agents Chemother* 64:e00251–20.

13. Balloni S, Locci P, Lumare A, Marinucci L. 2016. Cytotoxicity of three commercial mouthrinses on extracellular matrix metabolism and human gingival cell behaviour. *Toxicol In Vitro* 34:88-96.
14. Zhang N, Chen C, Melo MA, Bai YX, Cheng L, Xu HH. 2015. A novel protein-repellent dental composite containing 2-methacryloyloxyethyl phosphorylcholine. *Int J Oral Sci* 7:103-9.

August 2, 2022

Dr. Derong Yin
State Key Laboratory of Oral Diseases, National Clinical Research Center for Oral Diseases, West China Hospital of Stomatology, Sichuan University
Section 3 RenMin South Road
Chengdu
China

Re: Spectrum01578-22R1 (Caffeic Acid Phenethyl Ester (CAPE) Inhibits Cross-Kingdom Biofilm Formation of Streptococcus mutans and Candida albicans)

Dear Dr. Derong Yin:

Your manuscript has been accepted, and I am forwarding it to the ASM Journals Department for publication. You will be notified when your proofs are ready to be viewed.

Sincerely,

Xiaoyu Tang
Editor, Microbiology Spectrum
